# Composite branched and linear F-actin maximize myosin-induced membrane shape changes in a biomimetic cell model
Ryota Sakamoto [1,2] & Michael P. Murrell [1,2,3] ✉

The architecture of the actin cortex determines the generation and transmission of stresses, during key events from cell division to migration. However, its impact on myosin-induced cell shape changes remains unclear. Here, we reconstitute a minimal model of the actomyosin cortex with branched or linear F-actin architecture within giant unilamellar vesicles (GUVs, liposomes). Upon light activation of myosin, neither the branched nor linear F-actin architecture alone induces significant liposome shape changes. The branched F-actin network forms an integrated, membrane-bound "no-slip boundary" -like cortex that attenuates actomyosin contractility. By contrast, the linear F-actin network forms an unintegrated "slip boundary" -like cortex, where actin asters form without inducing membrane deformations. Notably, liposomes undergo significant deformations at an optimized balance of branched and linear F-actin networks. Our findings highlight the pivotal roles of branched F-actin in force transmission and linear F-actin in force generation to yield membrane shape changes.

Cell shape changes are induced by the contractile forces of the actin cytoskeleton beneath the cell plasma membranes. Myosin II-induced stresses propagate through the F-actin network branched or crosslinked by actin-associated proteins[1]. These forces are transmitted to the plasma membrane via linker proteins[2,3], driving various cellular processes such as cell division[4], blebbing[5,6], and migration[7–10].

The F-actin in the cell cortex is predominantly nucleated by two proteins: the Arp2/3 complex and formin mDia1[11]. Growing evidence suggests that the F-actin cortex architecture, comprising branched F-actin nucleated by the Arp2/3 complex[12] and linear F-actin nucleated by formins like mDia1[13], significantly influences cortex mechanical properties and force generation. For example, the stiffness of the cell cortex largely depends on the activity of mDia1 despite its minor fraction (~10%) of the mDia1-nucleated linear F-actin within the cortex[14]. Cell cortex tension is significantly decreased by perturbing mDia1 activity[15]. Moreover, mDia1 is essential for cell division in HeLa cells, but not Arp2/3[11], while formin-mediated F-actin nucleation dominates upon mitotic entry in *Drosophila* epithelial cells[16]. Thus, F-actin architecture plays pivotal roles in cortex mechanical properties and cell function.

To date, the role of F-actin architecture is acknowledged in terms of myosin force generation and its transmission within the F-actin network[17–19]. In vitro studies in two-dimensional open systems have demonstrated that the contractile force of myosin is attenuated within highly branched F-actin gels nucleated by Arp2/3, while linear mDia1

networks allow contraction[17]. However, in cellular contexts, actomyosin is confined within a μm-scale boundary enclosed by a soft, deformable cell membrane. In this case, the transmission of forces to the membrane is crucial for performing specific functions such as cell division and ameboid-type migration[4,7–10]. Nevertheless, the precise influence of distinct F-actin architectures on membrane shape changes, and how these architectures are finely tuned to shape diverse cellular behaviors, remain poorly understood.

Understanding the role of F-actin architecture in cells is often challenged by variable protein composition and biochemical signaling molecules, such as Rho GTPases, regulating the activity of F-actin nucleator and myosin[20–22]. To isolate the contribution of F-actin architecture to myosin force generation and membrane deformations, we assembled the actomyosin cortex within giant unilamellar vesicles (GUVs, liposomes)[23–32]. Myosin was light-activated after assembling the actin cortex with branched or linear F-actin architecture[33,34], enabling us to quantitatively characterize the impacts of F-actin architecture on membrane deformations. The branched F-actin cortex was formed using the Arp2/3 complex, while the linear F-actin cortex was formed using mDia1. Notably, significant membrane deformations were observed only when the branched and linear networks were mixed. Our findings highlight the importance of fine-tuning F-actin architecture by combining branched and linear F-actin, as observed in the cell cortex, providing mechanical insights into the coordination among F-actin architecture, myosin force generation, and cell shape changes.

[1]Department of Biomedical Engineering, Yale University, 10 Hillhouse Avenue, New Haven, CT, USA. [2]Systems Biology Institute, 850 West Campus Drive, West Haven, CT, USA. [3]Department of Physics, Yale University, 217 Prospect Street, New Haven, CT, USA. ✉e-mail: michael.murrell@yale.edu

## Results

### Light activation of myosin force generation within liposomes

First, we establish precise control over actomyosin contractility within GUVs. Purified actin, myosin, and actin-associated proteins were encapsulated into liposomes using the inverted emulsion method (Methods)[23,35]. The preparation of actomyosin-encapsulated liposomes takes ~30 min, during which myosin activity and F-actin polymerization were kept minimal by maintaining liposomes below 4 °C. F-actin polymerization was initiated at room temperature (R.T. ~25 °C), whereas it enabled actomyosin contraction as well (Supplementary Fig. 1). To investigate the impact of F-actin architecture on membrane deformations, myosin was inactivated by blebbistatin, a myosin ATPase activity inhibitor[36], before F-actin network assembly (Fig. 1a). After full assembly of the F-actin network, myosin was activated by inactivating blebbistatin using a 405 nm laser[33,34], initiating actomyosin network contraction (Fig. 1a–c, Movie S1).

To quantitatively characterize the contracting behavior of the light-activated actomyosin network, we performed Particle Image Velocimetry (PIV) to calculate the displacement of the actin network and strain fields (Method) (Fig. 1d). Comparable sizes of liposomes with diameter ~45–60 μm were analyzed to reduce variability (Supplementary Fig. 2). Strain variables and strain rates increased with higher myosin concentration, validating the light-activated actomyosin contraction in liposomes (Fig. 1e–h). Having established the light-activation system in liposomes, we then investigated the impact of actin cortex architecture on membrane deformations.

### Mixed F-actin cortex architecture enables membrane deformations

To assemble the actin cortex, we employed His-tagged F-actin nucleators anchored to the inner leaflet of lipid bilayer membranes containing 1,2-dioleoyl-sn-glycero-3-[(N-(5-amino-1-carboxypentyl)iminodiacetic acid) succinyl] (nickel salt) (18:1 DGS-NTA(Ni)) lipids (Methods)[23,30]. All His-mDia1, a linear F-actin nucleator, and His-VCA, a WASP fragment that activates the Arp2/3 complex-mediated branched F-actin assembly, were assumed to be membrane-bound[17,30,37]. Order of magnitude estimation demonstrated that there are sufficient membrane binding sites for all His-tagged proteins under our experimental conditions (Methods).

First, we used the branched F-actin nucleator, the Arp2/3 complex ([Arp2/3] = 25 nM). To localize the assembly of the branched F-actin network beneath the membrane, His-VCA, an activator of Arp2/3-mediated F-actin assembly, was localized to the membrane (Fig. 2a)[23,30]. F-actin was polymerized for ~10 min at R.T. (~25 °C) after liposome preparation. Subsequently, myosin was light-activated, and liposome shape changes were captured in timelapse images with 6 s intervals. Comparable sizes of liposomes with diameter ~35–55 μm were analyzed to reduce variability (Supplementary Fig. 2).

The branched F-actin cortex architecture was examined by focusing on the bottom surface of the liposomes, where the network can be observed within a single focal plane (Fig. 2b, top). The orientation director field of F-actin and the nematic order parameter, $S = \langle \cos 2\theta \rangle$, were calculated to distinguish the actin cortex architecture, where $\theta$ is the local angle of the director field, $\langle \cdot \rangle$ is the local average (Methods) (Fig. 2b, bottom). The mean

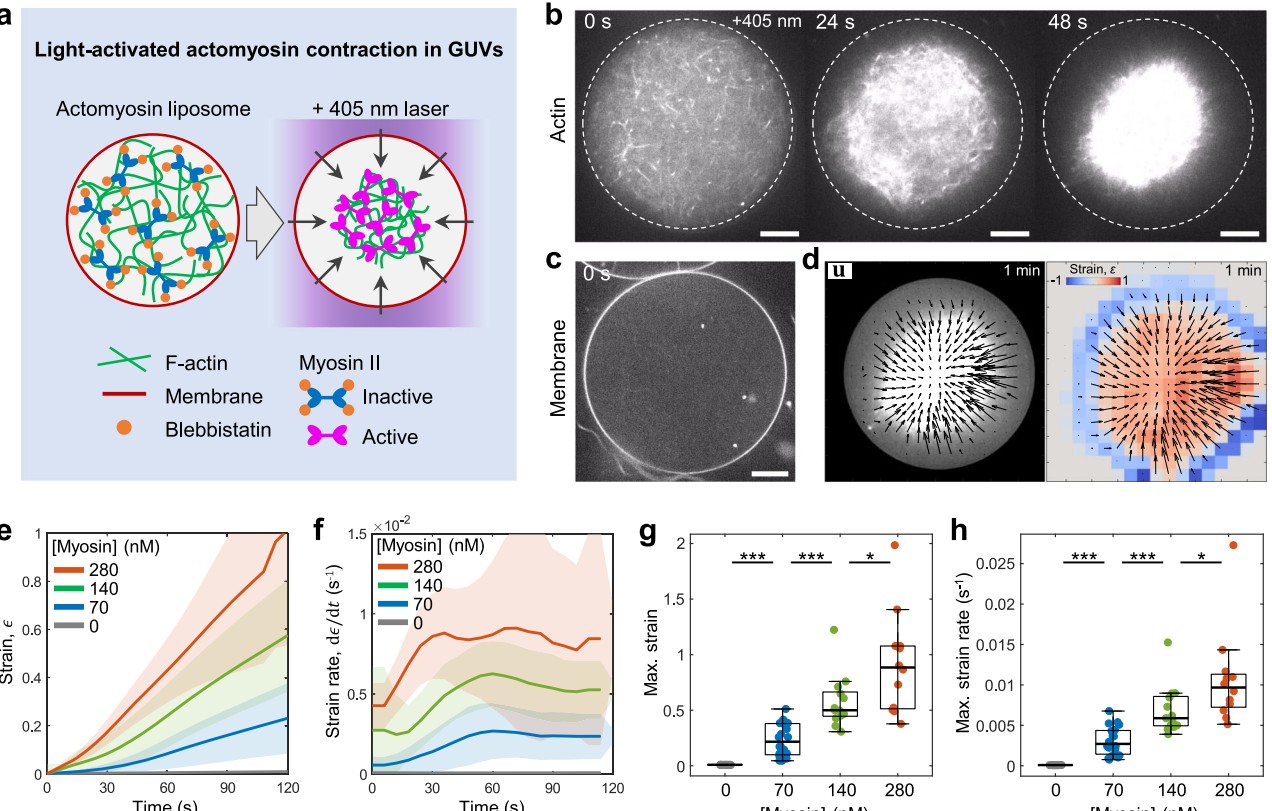

**Fig. 1 | Light-activated contraction of the actomyosin networks within liposomes.** **a** Schematic of the light-activation of actomyosin contractility within a liposome encapsulating purified actin and myosin. **b** Timelapse images showing the mid-plane of the light-activated contracting actomyosin network within a liposome. The dashed line represents the position of the membrane. **c** The snapshot showing the fluorescently labeled lipid bilayer membrane in a mid-plane. **d** Actomyosin network contraction with 280 nM myosin. Black arrows are the total displacement, $u$, over 1 min, and vector magnitudes are normalized by its maximum. Colormap represents local strain fields. Mean compressive strain (**e**) and strain rate (**f**) over time ($n = 12$ liposomes and $N = 2$ independent experiments in 0 nM; $n = 20$ and $N = 2$ in 70 nM; $n = 13$ and $N = 2$ in 140 nM; $n = 12$ and $N = 3$ in 280 nM). Maximum strain (**g**) and maximum strain rate (**h**) extracted from (**e**) at 120 s and (**f**) ($n = 12$ liposomes and $N = 2$ independent experiments in 0 nM; $n = 20$ and $N = 2$ in 70 nM; $n = 13$ and $N = 2$ in 140 nM; $n = 12$ and $N = 3$ in 280 nM). Curves are mean ± SD. * and *** represent $p < 0.05$ and $p < 0.001$, respectively. Scale bars, 10 μm.

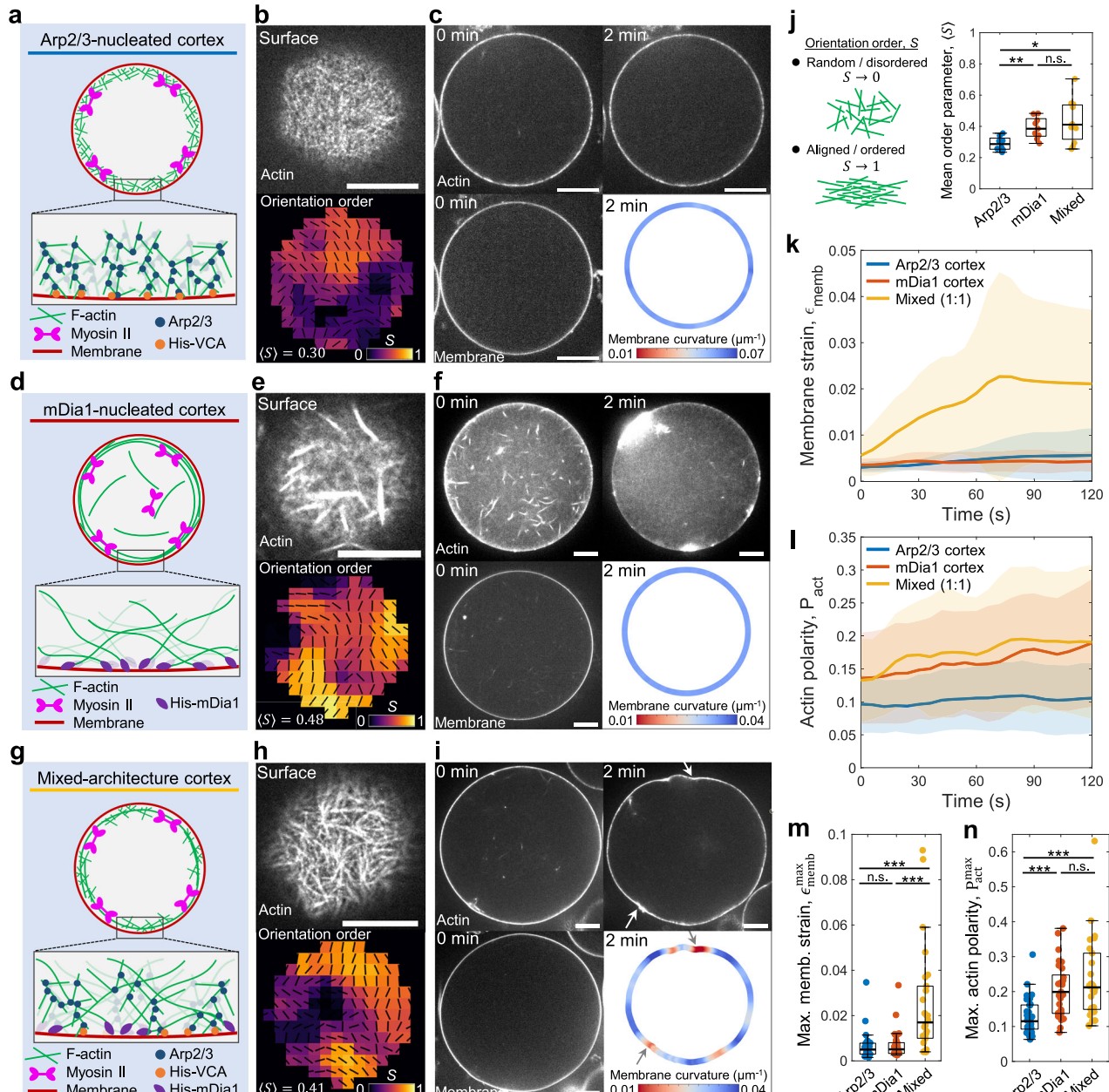

**Fig. 2 | Assembly of distinct actin cortex architectures within liposomes.**
**a** Schematic of the Arp2/3-nucleated branched F-actin cortex activated via His-VCA
([Arp2/3] = 25 nM). **b** A snapshot showing the surface of the Arp2/3 cortex liposome
before 405 nm illumination. The nematic order parameter, $S = \langle\cos 2\theta\rangle$, is calculated
from the director field (black dashes). **c** Snapshots showing the actin and membrane
before and post 405 nm illumination. Membrane curvature is color-coded at 2 min.
**d** Schematic of the mDia1-nucleated linear F-actin cortex ([mDia1] = 25 nM). **e** A
snapshot showing the surface of the mDia1 cortex liposome and nematic order para-
meter. **f** Snapshots showing the actin and membrane before and post 405 nm illumi-
nation. Membrane curvature is color-coded at 2 min. **g** Schematic of the mixed F-actin
architecture cortex at [mDia1]:[Arp2/3] = 1:1 ([mDia1] = 25 nM; [Arp2/3] = 25 nM).

**h** A snapshot showing the surface of the mixed cortex liposome and nematic order
parameter. **i** Snapshots showing the actin and membrane before and post 405 nm illu-
mination. White arrows indicate local deformations. Membrane curvature is color-coded
at 2 min. **j** Boxplot showing the mean order parameter ($n = 8$ liposomes and $N = 5$
independent experiments in Arp2/3; $n = 9$ and $N = 3$ in mDia1; $n = 11$ and $N = 2$ in
Mixed). Membrane strain (**k**) and actin polarity (**l**) over time ($n = 33$ and $N = 6$ in Arp2/3;
$n = 24$ and $N = 5$ in mDia1; $n = 26$ and $N = 2$ in Mixed). Maximum membrane strain (**m**)
and maximum actin polarity (**n**) extracted from (**k**) and (**l**) ($n = 33$ and $N = 6$ in Arp2/3;
$n = 24$ and $N = 5$ in mDia1; $n = 26$ and $N = 2$ in Mixed). Curves are mean ± SD. *, **, and
*** represents $p < 0.05$, $p < 0.01$, and $p < 0.001$, respectively. n.s. not significant. Scale
bars, 10 μm.

order parameter, $\langle S \rangle$, was calculated by averaging $S$ over the entire bottom
surface. The branched actin cortex exhibited a low mean nematic order
parameter, indicative of a disordered network ($\langle S \rangle = 0.29 \pm 0.04$, mean ±
SD) (Fig. 2j). We did not observe clear membrane deformation upon light
activation of the Arp2/3-nucleated cortex (Fig. 2c, Movies S2, S3). This
observation aligns with previous reports that Arp2/3-nucleated F-actin
networks attenuate contractility and prevent contraction of the F-actin
network in two-dimensional open systems[17].

The linear F-actin cortex was formed by localizing His-mDia1 beneath
the membrane ([mDia1] = 25 nM, where His-mDia1 concentration is
denoted as [mDia1] thereafter) (Fig. 2d). Profilin was included to facilitate
mDia1-mediated F-actin assembly (Methods). There was no His-VCA
present in the absence of Arp2/3. We observed linear F-actin localized
beneath the membrane, characterized by a high mean nematic order
parameter, indicating an ordered network ($\langle S \rangle = 0.39 \pm 0.07$) (Fig. 2e, j).
Since mDia1-nucleated F-actin could be released from the membrane-

anchored His-mDia1 after polymerization[37], non-localized F-actin was also present in the volume region of the liposomes (Fig. 2f). Notably, the mDia1-nucleated cortex rapidly contracted and formed asters beneath the membrane upon light-activation of myosin, as reported in two-dimensional open systems[17] (Fig. 2f, Movies S4, S5). However, significant membrane deformation was not observed (Fig. 2f, membrane curvature). This suggests that the linear F-actin network enables myosin force generation within the actin cortex, while the contractile forces are not effectively transmitted to the membrane due to the lack of connectivity within the F-actin network and between the cortex and membrane.

Next, we assembled a mixed architecture cortex composed of composite branched and linear F-actin networks, nucleated by a 1:1 mixture of Arp2/3 and mDia1 ([Arp2/3] = 25 nM; [mDia1] = 25 nM)) (Fig. 2g). We retained the Arp2/3 concentration the same as that in the Arp2/3 only network to maintain the influence of branched F-actin architecture. The mixed architecture cortex exhibited a dense linear filamentous network at the surface, with a higher mean nematic order parameter than the Arp2/3 cortex ($\langle S \rangle = 0.43 \pm 0.13$ in Mixed; $\langle S \rangle = 0.29 \pm 0.04$ in Arp2/3) (Fig. 2h, j). Strikingly, upon light-activation of myosin, the mixed architecture cortex liposomes exhibited significant membrane deformation with local curvature changes (Fig. 2i, Movies S6, S7). This membrane deformation was likely facilitated by an excess membrane of the slightly deflated liposome (Methods). Interestingly, some liposomes displayed polarized cortical actin flow (~12% of liposomes, Supplementary Figs. 3, 4, Movies S8, S9), reminiscent of cortical actin flow observed during cell migration[9]. These results suggest that mDia1-nucleated linear F-actin networks facilitate myosin force generation within the cortex, while the branched, membrane-linked Arp2/3-nucleated F-actin networks enable force transmission from the cortex to the membrane, resulting in significant membrane deformations.

We note that blebbistatin inactivation was performed both within and outside the focal plane using a confocal microscope (Methods). Deformations may appear to occur in one or two locations by chance (Movies S6, S7), likely due to initial heterogeneity in the distribution of actin and myosin. We confirmed membrane deformation outside the mid-plane after the light activation of the mid-plane (Supplementary Fig. 5). The maximum z-projection of the z-stack images showed multiple contracted aster-like structures on the surface of the post-light activated liposomes (Supplementary Fig. 5).

It is noteworthy that contraction dynamics ceased a few minutes after light activation in many cases. This aligns with well-established observations that actomyosin network contraction finishes after forming contracted asters/aggregates without observable dynamics in two-dimensional assays[17,33,38]. In cases where there is no significant actin turnover, it is commonly observed that contraction dynamics cease after a few minutes, eventually forming local contraction spots. Second, as discussed in the theoretical model below, deformation dynamics may cease when the energetic cost of deformation balances myosin-generated active stress. Third, the possibility of photodamage to myosin is unlikely since we observed persistent, slower contractions at lower myosin concentrations compared to high concentrations (Movie S1). Finally, ATP depletion is also unlikely, given the ATP hydrolysis rate of actin is ~4 µM min$^{-1}$ at the present actin concentration (6 µM)[19], ensuring the ATP concentration of ~5 mM within the liposome after actin polymerization (~10 min).

## Membrane strain and actin polarity analysis

To quantitatively characterize membrane deformation and actomyosin contraction, we introduce two metrics: membrane strain, $\epsilon_{memb}$, and actin polarity, $P_{act}$. Membrane strain $\epsilon_{memb}$ measures how much the liposome shape deviates from a sphere. Roundness of the liposome is quantified by the circularity metric, $C = 4\pi \times S/L^2$, where $S$ is the closed area and $L$ is the perimeter of the liposome. This metric is 1 for a sphere and decreases for deformed shapes. Membrane strain is defined as $\epsilon_{memb} \equiv 1\text{-}C$, with larger values indicating more significant deformation.

Actin polarity $P_{act}$ measures the heterogeneity of the spatial distribution of the actomyosin network within liposomes. First, the center-of-mass of the liposome, $\mathbf{r}_c = N^{-1}\Sigma\mathbf{r}_i$, is calculated from the membrane positions, $\mathbf{r}_i = (x_i, y_i), i = 1 \cdots N$. The actin intensity at each membrane position, $I_{act}(\mathbf{r}_i)$, is then measured. The center-of-mass of the actin distribution, $r_{act}$, is calculated as the actin intensity-weighted position of the membrane: $\mathbf{r}_{act} = \Sigma\mathbf{r}_iI_{act}(\mathbf{r}_i)/\Sigma I_{act}(\mathbf{r}_i)$. Finally, the actin polarity is defined as the deviation of the actin distribution from the center-of-mass of the liposome, normalized by the initial liposome radius, $R$: $P_{act} \equiv |\mathbf{r}_{act} - \mathbf{r}_c|/R$. Higher actin polarity indicates a more heterogeneous, localized actin distribution (e.g., asters).

Immediately after light-activation, the membrane strain of the mixed cortex rapidly increased over ~60 s, reached a plateau, while both the Arp2/3 cortex and the mDia1 cortex showed no significant change (Fig. 2k, m). This demonstrates that significant membrane deformation occurs only in the mixed architecture cortex liposomes. On the other hand, although the mDia1 cortex did not exhibit membrane deformation, its actin polarity increased as high as the mixed cortex ($P_{act}^{max}$ ~0.20 in mDia1; $P_{act}^{max}$ ~0.24 in Mixed) (Fig. 2l, n). Thus, mDia1-nucleated linear F-actin networks alone can facilitate myosin force generation within the network but are insufficient to effectively transmit contractile forces to the membrane. In contrast, the undistorted Arp2/3 cortex exhibited significantly lower actin polarity ($P_{act}^{max}$ ~0.13 in Arp2/3) (Fig. 2n), indicating that contractility is attenuated within the Arp2/3-nucleated branched cortex[17]. Together, the quantitative analysis of membrane strain and actin polarity suggests distinct roles of F-actin architectures in force generation, transmission, and membrane deformations.

## Mechanical characterization of membrane deformations

To understand how changes in actin cortex architecture yield changes in morphological deformation modes, we performed spectral correlation analysis on membrane contour shapes. The radial position of the membrane, $R(\theta, t)$, is measured from the center-of-mass of the liposome in the mid-plane (Fig. 3a). The amplitude of membrane deformation is defined as $u(\theta, t) = R(\theta, t) - \langle R(\theta, t) \rangle_\theta$, where $\langle \cdot \rangle_\theta$ is the average overall $\theta$. This analysis quantifies the angular distribution of membrane deformations (Fig. 3b–d). The maximum radial membrane deformation, $R_{max}/\langle R(\theta, t) \rangle_\theta$, was significantly higher in the mixed architecture cortex compared to the Arp2/3 cortex and mDia1 cortex (Fig. 3e), consistent with the membrane strain analysis.

The scaling exponent of the power spectrum of membrane deformation reflects the membrane mechanical properties. Specifically, the power spectrum of membrane deformation scales as $\langle |u(q)|^2 \rangle_t \sim (\kappa q^4 + \gamma q^2)^{-1}$, where $\kappa$, $\gamma$, and $q$ are the bending rigidity, surface tension, and wave number, respectively. A scaling exponent around $q^{-2}$ to $q^{-3}$ represents a tension-dominated regime, while a scaling exponent around $q^{-3}$ to $q^{-4}$ indicates an elasticity-dominated regime[39–41]. We calculated the power spectrum of membrane deformation from the amplitude of membrane deformation (Fig. 3f). The Arp2/3 cortex and mixed architecture cortex follow scaling close to $q^{-4}$, while the mDia1 cortex follows scaling close to $q^{-2}$ (Fig. 3g). This suggests that the branched Arp2/3 cortex behaves as an elastic sheet beneath the membrane, inducing elastic deformation. In contrast, the mDia1-nucleated linear F-actin cortex does not exhibit elastic contribution to the membrane, but membrane tension dominates. Thus, the mDia1 cortex does not induce significant membrane deformation.

Finally, the characteristic size of deformation, $\theta_c$, is calculated from the smallest $\theta$ that satisfies the autocorrelation of the membrane amplitude, $\langle u(\theta + \Delta\theta, t')u(\theta, t') \rangle_\theta = 0$, where $t'$ is the time point at which $\theta_c$ takes the smallest value (Methods). The autocorrelation of the Arp2/3 cortex showed a single negative peak corresponding to ellipsoidal deformation, whereas the mixed architecture cortex typically had more than two negative peaks, exhibiting more complex deformation with higher modes (Fig. 3h). Since the Arp2/3 cortex did not show complex deformation, its deformation size was larger than that of the mixed architecture cortex ($\theta_c \sim 32°$ in Arp2/3

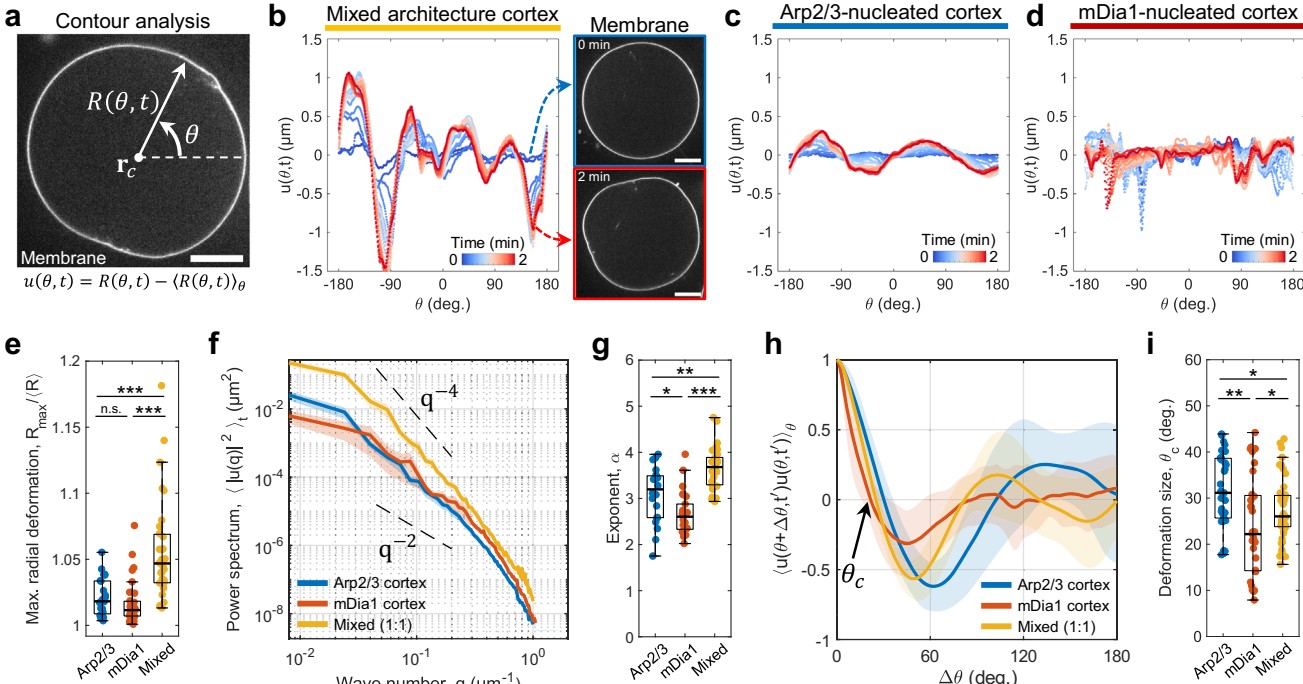

**Fig. 3 | Contour shape analysis on deformed membranes. a** Snapshots showing the definition of radial membrane position $R(\theta, t)$ and deformation amplitude $u(\theta, t)$. **b** Time evolution of the deformation amplitude in mixed architecture cortex. Time is color-coded. The right snapshots show the mid-plane image of the membrane at 0 min and 2 min. **c** Time evolution of the deformation amplitude in Arp2/3-nucleated cortex. **d** Time evolution of the deformation amplitude in mDia1-nucleated cortex. **e** Boxplot showing maximum radial deformation ($n = 24$ liposomes and $N = 2$ independent experiments in Arp2/3; $n = 31$ and $N = 2$ in mDia1; $n = 31$ and $N = 3$ in Mixed). **f** Power spectrum of membrane deformation with different F-actin cortex architecture. The power spectrum is averaged over 20 frames for each liposome, which is averaged for all liposomes. Curves are mean ± SEM.

Dotted lines are the eye guide for $q^{-4}$ and $q^{-2}$ ($n = 14$ and $N = 2$ in Arp2/3; $n = 13$ and $N = 2$ in mDia1; $n = 15$ and $N = 2$ in Mixed). **g** Boxplot showing scaling exponent $\alpha$. Scaling exponent is extracted by fitting the power spectrum by a function $q^{-a}$ for all $q$ ($n = 20$ and $N = 2$ in Arp2/3; $n = 26$ and $N = 2$ in mDia1; $n = 30$ and $N = 3$ in Mixed). **h** Autocorrelation of the deformation amplitude. $\theta_c$ is the deformation size calculated at the smallest $\theta$ at which autocorrelation becomes 0. The autocorrelation at the time frame $t'$ was shown at which $\theta_c$ takes the smallest value. Curves are mean ± SD ($n = 14$ and $N = 2$ in Arp2/3; $n = 13$ and $N = 2$ in mDia1; $n = 15$ and $N = 2$ in Mixed). **i** Boxplot showing the deformation size ($n = 33$ and $N = 3$ in Arp2/3; $n = 32$ and $N = 3$ in mDia1; $n = 49$ and $N = 3$ in Mixed). *, **, and *** represent $p < 0.05$, $p < 0.01$ and $p < 0.001$, respectively. n.s. not significant. Scale bars, 10 μm.

cortex; $\theta_c \sim 28°$ in Mixed cortex) (Fig. 3i). On the other hand, the mDia1 cortex exhibited the smallest deformation size ($\theta_c \sim 24°$), caused by locally contracted small actin asters beneath the membrane. Together, spectrum and correlation analysis demonstrate that F-actin architecture can alter the mechanical response to myosin force generation and change membrane deformation modes, providing mechanical insights into F-actin architecture-dependent membrane deformation behaviors.

We estimate the cortex tension and mechanical work performed to deform the membrane using a theoretical model based on the energy cost of deformation as developed in previous studies[41,42]. In brief, to slightly deform a portion of the spherical liposome with a depth $\delta$, actomyosin contractility must overcome the energy cost associated with deforming the elastic cortex and membrane. The elastic energy comprises bending energy $F_{bend}$ and stretching energy $F_{stretch}$ due to deformation. Additionally, the change in the membrane surface area contributes to the energy cost $F_{memb}$. At the onset of deformation, the work done by contraction, $W_{contract}$, is approximately equal to the energy cost of deformation: $W_{contract} \sim F_{bend} + F_{stretch} + F_{memb}$.

Using typical values for the liposome size $R \sim 20$ μm, cortex thickness $h = 0.29 \pm 0.09$ μm (Supplementary Figs. 6, 7), deformation depth $\delta$ estimated from deformation amplitude as $\delta \sim R_{max} - \langle R(\theta, t) \rangle_\theta \sim 1.1$ μm (Fig. 3e), deformation size $\theta_c \sim 28°$ (in correlation length, $\xi_c \sim R\theta_c \sim 9.7$ μm) (Fig. 3i), elastic modulus of the Arp2/3-branched F-actin network $E \sim 4 \times 10^3$ Pa[43,44], and membrane tension $\gamma_m \sim 10^{-7}$ N m$^{-1}$ [45,46], we estimate $F_{bend} \sim Eh^3\delta^2/\xi_c^2 \sim 1.2 \times 10^{-18}$ J, $F_{stretch} \sim Eh\xi_c^2\delta^2/R^2(1 + \xi_c^2/R^2) \sim 8.4 \times 10^{-17}$ J, $F_{memb} \sim \gamma_m \xi_c^2\delta/R \sim 10^{-19}$ J. The energy cost of stretching contributes the most significantly[41]. The cortex tension $\gamma_c$ is estimated from the mechanical work, $W_{contract} \sim h\sigma_{act}\xi_c^2\delta/R$, where $\sigma_{act}$ is the tangential active stress, and with the relationship, $\gamma_c = h\sigma_{act}$[47,48], yielding

$\gamma_c \sim 1.6 \times 10^{-5}$ N m$^{-1}$, which is one order of magnitude smaller than that of living cells[47].

Interestingly, polarized cortical actin flow was observed in liposomes with high actin polarity ($P_{act} > 0.3$) (Supplementary Figs. 3, 4, Movies S8, S9), reminiscent of cortical flows during cell migration[9]. In this case, we estimate the cortex tension $\gamma_c$ from the force balance between the active stress $\sigma_{act}$ and membrane-induced friction stress $f_{fric}$ using the relationships, $\gamma_c = h\sigma_{act}$[47,48] and $f_{fric} = \zeta v_{act}$[24], where $h$ is the cortex thickness, $v_{act}$ is the actin flow speed, and $\zeta = \eta_m N_{link}/(4\pi R^2)$ is the friction coefficient between the cortex and the membrane. $\eta_m$ and $N_{link}$ are the membrane viscosity and the total number of cortex-to-membrane links, respectively. Using the measured values $v_{act} = 13.3 \pm 4.0$ μm min$^{-1}$ (Supplementary Fig. 3) and $h = 0.29 \pm 0.09$ μm (Supplementary Figs. 6, 7), and typical literature values $\eta_m \sim 10^{-5}$ Pa s m[29,35], and $N_{link} \sim 5 \times 10^5$ (assuming all Arp2/3 is bound to the membrane-bound His-VCA sites in a liposome with $R = 20$ μm[24,29]), the cortex tension is estimated to be $\gamma_c \sim 6.4 \times 10^{-5}$ N m$^{-1}$, comparable to the cortex tension estimated from the energy cost of membrane deformation.

It should be noted that small liposomes ($R < 10$ μm) tended to exhibit minor membrane deformation (Supplementary Fig. 8). For example, the membrane strain of liposomes with $R \simeq 10$ μm was significantly smaller than that of liposomes larger than $R \simeq 20$ μm (Supplementary Fig. 8). This observation could be explained by the bending energy cost for smaller liposomes being much higher than that for larger ones. Based on the mechanical energy of deformation described above, the ratio of the bending energy cost to the stretching energy cost of the actin cortex scales with radius (curvature), $F_{bend}/F_{stretch} \sim h^2R^{-2}$, when the deformation size (correlation length) is comparable to the liposome size, $\xi_c \sim R$[41]. Thus, for smaller liposomes, the bending energy cost dominates, making membrane

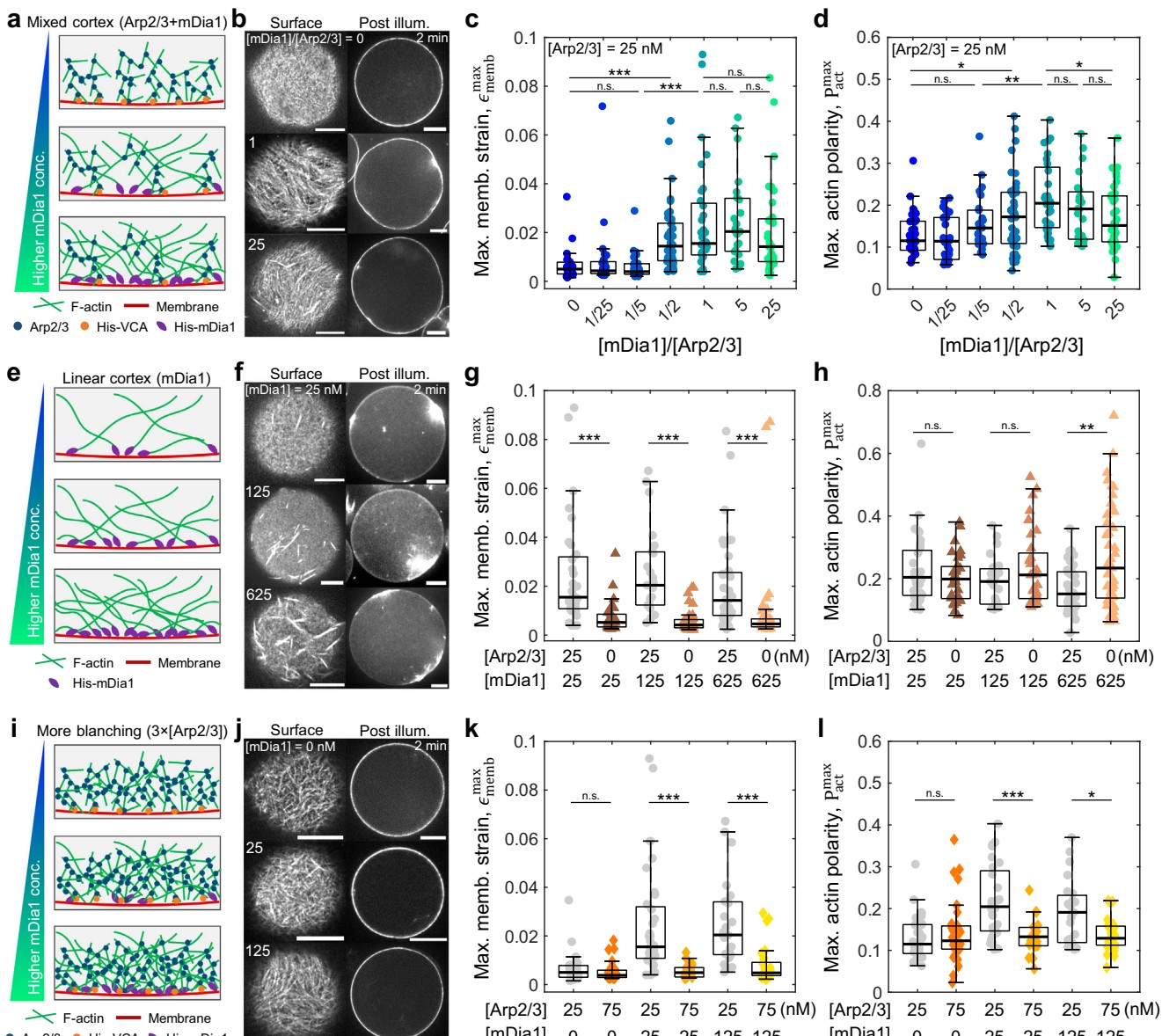

**Fig. 4 | A fine-tuning of the mixed F-actin architecture facilitates membrane deformation. a** Schematic of the mixed F-actin cortex with increasing mDia1 concentration. **b** Snapshots of the surface of the liposome and the mid-plane of the actin cortex at post-illumination (after 2 min) for different mDia1 to Arp2/3 ratio. Boxplots showing maximum membrane strain (**c**) and maximum actin polarity (**d**) at varied mDia1 to Arp2/3 ratio ($n = 33$ liposomes and $N = 2$ independent experiments in $[mDia1]/[Arp2/3] = 0$; $n = 28$ and $N = 2$ in 1/25; $n = 26$ and $N = 2$ in 1/5; $n = 41$ and $N = 2$ in 1/2; $n = 32$ and $N = 3$ in 1; $n = 23$ and $N = 2$ in 5; $n = 36$ and $N = 2$ in 25). **e** Schematic of the mDia1-nucleated linear F-actin cortex with increasing mDia1 concentration. **f** Snapshots of the surface of the liposome and the mid-plane of the actin cortex at post-illumination (after 2 min) for different mDia1

concentrations. Maximum membrane strain (**g**) and maximum actin polarity (**h**) at varied mDia1 concentration ($n = 31$ and $N = 2$ in $[mDia1] = 25$ nM; $n = 36$ and $N = 2$ in 125 nM; $n = 49$ and $N = 2$ in 625 nM). **i** Schematic of the Arp2/3-nucleated branched F-actin cortex at three times higher Arp2/3 concentration than the control condition in (**a**) with increasing mDia1 concentration. **j** Snapshots of the surface of the liposome and the mid-plane of the actin cortex at post-illumination (after 2 min) for different mDia1 concentrations. Maximum membrane strain (**k**) and maximum actin polarity (**l**) at varied mDia1 concentration ($n = 31$ and $N = 3$ in $[mDia1] = 0$ nM; $n = 22$ and $N = 2$ in 25 nM; $n = 29$ and $N = 3$ in 125 nM). *, **, *** represent $p < 0.05$, $p < 0.01$, and $p < 0.001$, respectively. n.s. not significant. Scale bars, 10 μm.

deformation difficult. Additionally, the smaller liposomes would have less membrane-localized myosin due to a larger surface-to-volume ratio, which may decrease the net active stress applied to the actin cortex. Together, these factors may limit deformation in smaller liposomes. We analyzed liposomes larger than $R \simeq 20 \mu m$ to ensure significant membrane deformation for deformation analysis.

## Fine-tuning composite architectures facilitates membrane deformation

To validate the influence of different F-actin architectures on the extent of shape changes, we systematically varied the molar ratio of mDia1 and Arp2/3

while keeping the Arp2/3 concentration constant at $[Arp2/3] = 25$ nM (Fig. 4a). Notably, significant membrane deformation was observed at $[mDia1]/[Arp2/3] > 0.5$ ($\epsilon_{memb}^{max} \sim 1.8 \times 10^{-2}$), whereas the membrane strain was comparable to the Arp2/3 cortex at lower ratios ($\epsilon_{memb}^{max} \sim 0.6 \times 10^{-2}$) (Fig. 4b, c). Actin polarity followed a similar trend as membrane strain (Fig. 4d). The fluorescence intensity of the actin cortex increased with higher His-mDia1 concentration, peaking around $[mDia1]/[Arp2/3] = 1$ (Supplementary Fig. 6). This suggests an optimal mDia1 concentration for maximizing linear F-actin localization beneath the membrane in the mixed architecture cortex. Beyond this concentration, F-actin released from membrane-bound mDia1 may dominate the volume space within the liposomes.

In the absence of Arp2/3, membrane strain remained small even with a 25 times higher His-mDia1 concentration ($\epsilon_{\mathrm{memb}}^{\max} \sim 0.9 \times 10^{-2}$) (Fig. 4e-g). Thus, mDia1-nucleated F-actin cortex alone cannot deform the membrane even at high mDia1 concentrations. In contrast, actin polarity is as high as in the significantly deformed mixed architecture cortex ($P_{\mathrm{act}}^{\max} \sim 0.3$) (Fig. 4h). This suggests that the mDia1-nucleated linear F-actin facilitates myosin force generation. We confirmed the contribution of mDia1 to the membrane-localized F-actin polymerization by the results showing an increase in both actin cortex intensity and its surface-to-volume ratio at higher mDia1 concentrations (Supplementary Fig. 9). It should be noted that the membrane deformation of the composite branched and linear network is not due to the increased anchor density via His-mDia1 (Figs. 2 and 3) since increasing the His-mDia1 concentration 5–25 times did not induce membrane deformation for mDia1-only liposomes. Thus, the deformation is not caused by the increased anchoring sites but by the influence of the composite F-actin architecture.

Increasing the Arp2/3 concentration from [Arp2/3] = 25 to 75 nM (i.e., 3×[Arp2/3]) in the mixed architecture cortex resulted in significantly smaller membrane strain, indicating that a highly dense branched actin gel may attenuate contractility and suppress membrane deformation[17] (Fig. 4i–k). Actin polarity remained as small as in the [mDia1] = 0 nM even at the higher mDia1 concentrations ($P_{\mathrm{act}}^{\max} \sim 0.14$), indicating that denser branching suppresses myosin force generation (Fig. 4l). Denser branching mediated by a higher Arp2/3 concentration is known to increase the elasticity and the bending rigidity of the cortex[42,44,49], which may also contribute to negligibly small membrane deformation and actomyosin contraction at high Arp2/3 conditions. We confirmed that the actin cortex intensity increased with higher Arp2/3 concentration (Supplementary Fig. 9). Although a higher His-VCA concentration resulted in a higher actin cortex intensity, membrane strain was comparable to the control condition, indicating that stronger actin-membrane links may facilitate force transmission and membrane deformation even for a denser actin cortex (Supplementary Fig. 10). Together, these results further validate the hypothesis that mDia1-nucleated linear F-actin is responsible for force generation within the cortex, while Arp2/3-nucleated branched F-actin transmits contractility from the cortex to the membrane, highlighting the importance of fine-tuning the F-actin cortex architecture to achieve significant membrane deformations.

We tested different profilin concentrations and observed that F-actin was not significantly localized beneath the membrane at higher profilin concentrations, resulting in a low surface-to-volume actin intensity ratio ($I_S/I_V \sim 1.8$, Supplementary Fig. 11). This might be attributed to the combined effects of profilin. Previous studies have shown a biphasic bell-shaped dependence of the barbed end elongation rate of formin on profilin concentration[50,51]. This dependency arises from the competing effects of accelerating the barbed-end elongation rate while excess profilin competes with profilin-actin for binding to formin, thereby inhibiting nucleation[50,52]. Particularly for mDia1, the elongation rate increases with profilin concentration when the profilin-to-actin ratio is [Profilin]/[Actin] < 2.0[50]. Since we used a profilin-to-actin ratio at [Profilin]/[Actin] < 0.1, the observed bulkier F-actin distribution at higher profilin concentration likely stems from the enhanced elongation rate induced by a higher profilin concentration (Supplementary Fig. 11). Moreover, profilin could potentially decrease the number of Arp2/3-mediated branching beneath the membrane[53]. Additionally, the limited availability of actin resources for Arp2/3 in high profilin concentration may shift the dominance toward mDia1-nucleation[54]. These effects may contribute to the presence of F-actin both in the volume and at the cortex, resembling to mDia1 cortex liposomes. Since our interest lies in the deformation induced by the actin cortex rather than volume network contraction-induced shape changes, we restricted our analysis to the profilin concentration regime where F-actin is well-localized beneath the membrane, forming the actin cortex with a high surface-to-volume actin intensity ratio ($I_S/I_V \sim 7.7$, Supplementary Fig. 11).

### Actin cortex architecture determines membrane deformations
To understand the overall impact of F-actin architecture on membrane deformations, we mapped the two major experimental readouts, actin polarity (i.e., force generation within the network) and membrane strain (i.e., force transmission from the cortex to the membrane), in phase space (Fig. 5a, b). When the Arp2/3-nucleated branched F-actin predominated, the actin cortex exhibited small actin polarity and negligible membrane deformation ($\epsilon_{\mathrm{memb}}^{\max} \to 0$, $P_{\mathrm{act}}^{\max} \to 0$) (Fig. 5a, diamonds, and 5b). Conversely, in the absence of Arp2/3, a higher concentration of mDia1 increased actin polarity, indicating that force generation was facilitated by the mDia1-nucleated linear F-actin ($\epsilon_{\mathrm{memb}}^{\max} \to 0$, $P_{\mathrm{act}}^{\max} \to 1$) (Fig. 5a, triangles, and 5b). However, membrane deformation was not observed for the mDia1 cortex. When these two distinct F-actin architectures were mixed, both actomyosin contraction and membrane deformation were observed at [mDia1]/[Arp2/3] > 0.5 ($\epsilon_{\mathrm{memb}}^{\max} \to 1$, $P_{\mathrm{act}}^{\max} \to 1$) (Fig. 5a, circles, and 5b).

Overall, these results underscore the unique mechanical roles of the F-actin nucleators (Fig. 5c). In the Arp2/3-nucleated cortex, the combination of Arp2/3-mediated branching and actin-membrane links via His-VCA forms a no-slip boundary-like cortex-membrane composite (Fig. 5c, left). In this case, the densely branched network dampens contractility and hinders membrane deformation. By contrast, the mDia1-nucleated linear cortex facilitates contraction (Fig. 5c, right). However, as there is no branching or crosslinking within the mDia1-nucleated linear F-actin network, unlike the Arp2/3 cortex, the adjacent membrane acts as a fluid-like slip boundary, and the membrane remains undistorted. In the composite branched and linear F-actin networks, contractile forces within the actin cortex can be transmitted to the membrane via branches and actin-membrane links, enabling significant membrane deformations (Fig. 5c, bottom). Thus, our findings highlight the importance of fine-tuning the composite branched and linear cortex architecture to induce membrane shape changes, unveiling the cooperative impact of distinct F-actin architectures on force generation, transmission, and membrane deformations.

## Discussion
We developed a biomimetic model cortex coupled to a lipid bilayer membrane to investigate the mechanical roles of F-actin architecture on membrane deformations. Light activation of actomyosin contractility and precise assembly of the actin cortex with specific F-actin architecture using Arp2/3 and mDia1 enabled quantitative characterization. We found that the linear F-actin cortex allows contractile force generation within the cortex, while the branched F-actin cortex acts as a scaffold to transmit contractility from the cortex to the membrane. Strikingly, neither of the F-actin cortex architectures alone is capable of inducing membrane deformations; instead, a composite F-actin architecture is essential for significant membrane deformation. Together, this study unveiled the mechanical roles of the F-actin architecture formed by Arp2/3 and mDia1, highlighting the importance of fine-tuning the composite branched and linear F-actin cortex architecture for force generation, transmission, and membrane deformations.

The Arp2/3 cortex may prevent actomyosin contraction in multiple ways, functioning as an effective no-slip boundary condition as referred to in this study. Arp2/3-only networks could hinder myosin force generation by limiting myosin motion through dense branching[17]. Previous numerical studies suggested that myosin motion is confined and inhibited in a highly connected network[17,55]. Moreover, when stress is applied to crosslinked or branched networks, crosslinker unbinding or debranching, as well as F-actin severing, induces stress relaxation[56,57]. Membrane-to-cortex attachment also contributes to filament severing[38]. Thus, in a highly branched Arp2/3 cortex, myosin motion is inhibited, and debranching and severing of the F-actin network could potentially relax accumulated stress, potentially contributing to the suppression of force generation and membrane deformation in the Arp2/3-nucleated branched cortex[17]. Additionally, the viscoelasticity of branched actin networks could hinder clustering and resulting deformation[17,44]. Given that actin turnover rates are linked to the viscoelastic properties of the actin cortex[2], exploring how varying turnover rates influence actomyosin contraction dynamics and membrane deformation, such as blebbing, would also be an intriguing avenue for future research.

**Fig. 5 | F-actin architecture determines force generation, transmission, and membrane deformations. a** A phase diagram of the maximum membrane strain and the maximum actin polarity for different F-actin cortex architectures. The data points with error bars represent mean ± SD (In mixed, $n = 33$ liposomes and $N = 2$ independent experiments in [mDia1]/[Arp2/3] = 0; $n = 28$ and $N = 2$ in 1/25; $n = 26$ and $N = 2$ in 1/5; $n = 41$ and $N = 2$ in 1/2; $n = 32$ and $N = 3$ in 1; $n = 23$ and $N = 2$ in 5; n = 36 and $N = 2$ in 25. In mDia1 only, $n = 31$ and $N = 2$ in [mDia1]=25 nM; $n = 36$ and $N = 2$ in 125 nM; $n = 49$ and $N = 2$ in 625 nM. In 3 × [Arp2/3], $n = 31$ and N = 3 in [mDia1]=0 nM; $n = 22$ and $N = 2$ in 25 nM; $n = 29$ and $N = 3$ in 125 nM). **b** Schematic represents the F-actin architectural control of membrane deformations shown in (**a**). **c** Schematic summarizing the F-actin architectural control of force generation, transmission, and membrane deformations. The Arp2/3-nucleated cortex branches the F-actin network and forms membrane-to-cortex links, which provide a no-slip boundary-like condition. However, the contractility is attenuated within the highly branched actin gel. In contrast, the mDia1-nucleated cortex allows force generation within the network, while the unbranched nature of the network results in the slip boundary-like condition without membrane deformation. When both the Arp2/3-nucleated branch and mDia1-nucleated linear filaments coexist, contractility within the F-actin network is transmitted to the adjacent membrane, inducing a significant membrane deformation.

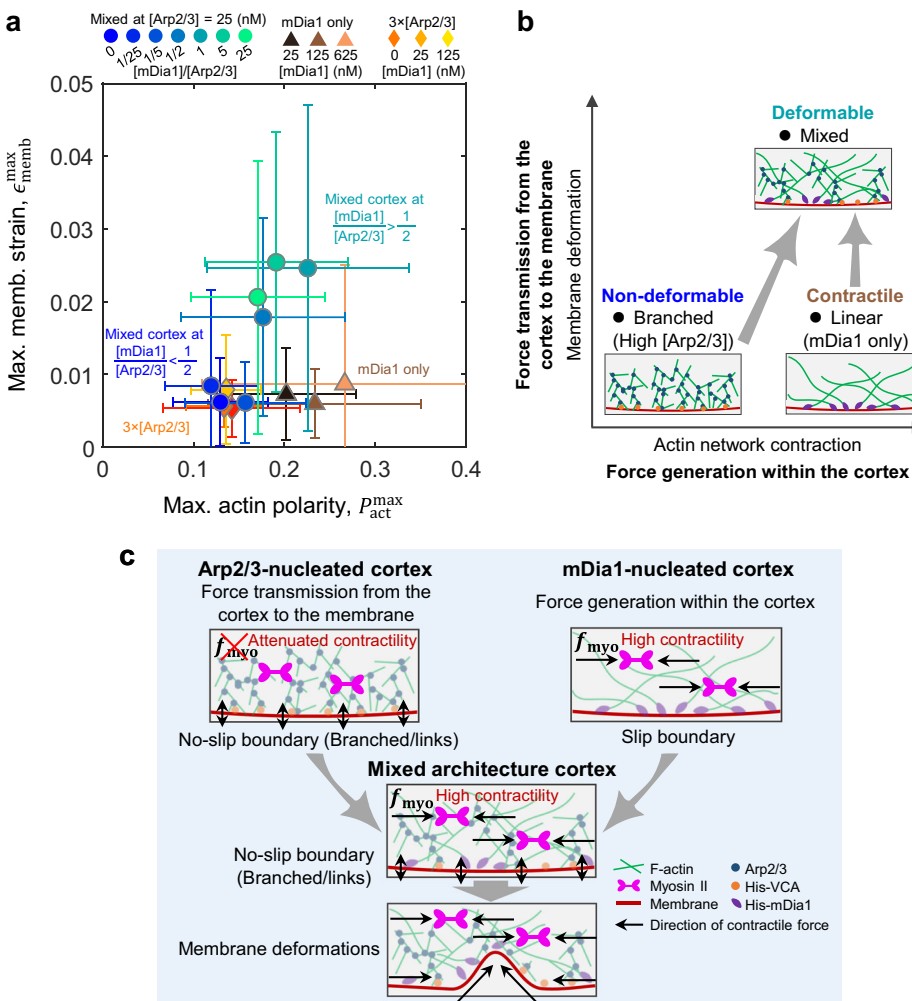

By contrast, within the mDia1-nucleated linear F-actin network, devoid of branching or crosslinking, actomyosin contractility induces an aster-like structure without significant membrane deformation. It remains unknown whether mDia1 unbinds from barbed ends as myosin motors pull on filaments, or if mDia1 is pulled together with the filament toward the myosin-induced asters. Contracting polar-aster-like structures may be generated by the cooperation of myosin contraction and formin-based actin nucleation as observed in cells[58]. Additionally, the force-sensing property of mDia1 may influence the contractile behavior during aster formation[59,60].

Within the composite branched and linear F-actin networks, long F-actin may span over multiple myosin motors. In this scenario, the debranching-based stress relaxation might be partially suppressed, as motors potentially enhance the crosslinking by acting as tentative cross-linkers between linear long F-actin networks[55,56]. Additionally, the mixed architecture network may facilitate force transmission due to the linear network branched via Arp2/3, effectively enabling force percolation[34,61,62]. Although estimating filament length within liposomes is challenging, single-molecule experiments and theoretical modeling studies have suggested that mDia1-nucleated filament length is ~10 times longer than Arp2/3-nucleated filament length in the HeLa cell cortex, and filament length was insensitive to the fraction of Arp2/3 and mDia1[14]. Thus, in the mixed F-actin architecture in our study, the length of F-actin elongated by mDia1 could be an order of magnitude longer than that of F-actin with free elongating barbed ends, potentially leading to some percolations. Therefore, the composite branched and linear F-actin network may represent the optimal architecture for force generation and transmission, facilitating membrane deformation. It is noteworthy that VCA may compete with formins for barbed ends through an interference mechanism[63]. Investigation of the

impact of a composite branched and linear F-actin network in agent-based simulations coupled with membrane deformation would provide pivotal insights into the mechanism of force transmission, integrating molecular-scale inputs to cell-scale outputs, in future studies.

The cell cortex comprises a mixed actin network nucleated by the Arp2/3 complex and formin mDia1[11], with ~10% of mDia1-nucleated linear F-actin and ~80% of Arp2/3-nucleated branched F-actin[14]. Recent studies have shown that mDia1 is essential for cell division in HeLa cells[11], and siRNA-induced depletion of mDia1 decreased cortex tension[14]. However, it remained unclear whether the cortex tension attenuation is attributable to the influence of the F-actin architecture of the Arp2/3-branched cortex or other actin-associated proteins, such as actin crosslinkers, capping, or severing proteins. In our study, we isolated the contribution of the branched F-actin architecture using in vitro reconstitution, revealing that the branched F-actin architecture alone effectively attenuates contractility and suppresses membrane deformation. Furthermore, we demonstrated that mDia1-nucleated linear F-actin architecture alone is insufficient to drive membrane deformation while fine-tuning the mixed F-actin architecture branched by Arp2/3 is essential to achieve significant membrane deformations. It is noteworthy that formin-mediated linear F-actin polymerization, but not Arp2/3, is crucial for retaining cortical flows during cell migration of primordial germ cells in vivo[64], where linear F-actin could be favored to facilitate flows without inducing shape changes. While direct comparisons with cellular situations are challenging due to the presence of actin-crosslinkers and membrane-to-cortex linkers, we suggest that F-actin architectures, relying solely on nucleators, may play predominant roles in force generation, transmission, and membrane deformation in cells.

To date, the reconstituted actomyosin cortex has been extensively studied in two-dimensional open systems[17,18], while its impact on three-dimensional membrane deformation remains elusive. Previous studies of actomyosin-encapsulated GUVs have reported a correlation between actomyosin contractility and membrane deformations[31,32,65]. However, the lack of spatial and temporal control of myosin activity has obscured the causal link between actomyosin contractility and membrane deformations. We addressed this limitation by establishing a light-activation system within actomyosin-encapsulated liposomes. This system provides a robust and versatile platform for investigating the impact of actomyosin contractility on membrane deformations, enabling us to quantitatively characterize the impact of F-actin nucleators[17], capping proteins[66], and crosslinkers[67] on the contractile behaviors of the actomyosin network coupled to membrane deformations.

Local activation of myosin would be a significant future direction[33,34], mimicking the intracellular signaling that triggers local myosin activation during cell division[4] and migration[20]. It is noteworthy that cortical actin nucleation shifts from an Arp2/3- to a formin-dominated regime upon mitotic entry in *Drosophila* epithelial cells[16]. Moreover, Arp2/3-mediated branched F-actin architecture is observed during adherent lamellipodia-based migration of fish keratocytes[68,69], whereas formin-mediated linear actin polymerization is essential to retain cortical flows during ameboid-type migration of primordial germ cells in vivo[64]. Interestingly, we observed cortical actin flows in mixed architecture cortex liposomes (Supplementary Figs. 3, 4 and Movies S8, S9), reminiscent of cortical flows during cell migration[9]. The in vitro reconstitution of these F-actin architecture-dependent cell division and migration mode-switching phenomena poses important future challenges.

## Methods

### Lipids preparation
Liposomes are formed with a combination of L-α-phosphatidylcholine from egg yolk (EPC) at 55.5% (840051; Avanti), cholesterol at 36% (ovine wool) (110796; Avanti), and 1,2-dioleoyl-sn-glycero-3- [n(5-amino-1-carboxypentyl) iminodiacetic acidsuccinyl nickel salt at 6.5% (18:1 DGS-NTA(Ni)) (790404; Avanti), 1,2-distearoyl-sn-glycero-3-phosphoethanolamine-N-[methoxy(polyethylene glycol)-2000] at 1% (18:0 PEG2000 PE) (880120; Avanti), and 1-palmitoyl-2-(dipyrrometheneboron difluoride) undecanoyl-sn-glycero-3-phosphocholine (TopFluor® PC) (810281; Avanti) at 1%.

### Buffers
Buffers were prepared by following the method described in previous studies[17,30,37,38]. The inner actin polymerization (IP) buffer contains 2 mM $CaCl_2$, 5 mM $MgCl_2$, 10 mM HEPES (pH 7.6), 0.8 mM DTT, 5 mM ATP, 50 mg ml$^{-1}$ dextran, and 175 mM sucrose. Myosin was kept inactivated by 68 μM blebbistatin (B0560; Sigma–Aldrich)[33,34]. ATP is regenerated through (PK) and phosphoenolpyruvate (PEP)-based ATP regeneration system used at 1 mM PEP (10108294001; Roche) and 10 U mL$^{-1}$ PK (P9136; Sigma–Aldrich). The outer buffer (OB) contains 0.2 mM $CaCl_2$, 0.2 mM $MgCl_2$, 10 mM HEPES (pH 7.6), 6 mM DTT, 2 mM ATP, 380 mM glucose, 0.01 mg/ml caseins. The osmolarity of the OB is adjusted with glucose such that the osmotic pressure difference between inner and OB is ~20–60 mOsm. Osmolarity of the OB is chosen to be slightly higher than the inner buffer to make the liposome slightly deflated, which enables membrane deformation. The OB is ~355 mOsm. The final KCl concentration was kept at 50 mM by adding additional amount according to the myosin concentration.

The storage buffer for actin (G-buffer) contains 2 mM Tris-HCl (pH 8.0) 0.1 mM $CaCl_2$, 0.2 mM ATP, and 0.5 mM DTT, and 22 μM actin. The storage buffer for myosin (MSB) contains 4.5 M KCl, 0.1 M HEPES (pH 7.0), and 21 μM myosin. The spin-down buffer (SDB) contains 2 μL phalloidin, 4.2 μL 238 μM actin, 40 μL IP buffer, 15 μL myosin (in MSB), 4.3 μL of 4.6 M KCl. Myosin is centrifuged in the presence of polymerized actin to isolate the catalytically active myosin dimers before use. Briefly, actin is

polymerized for 1 h at 4 °C in high salt (IP buffer +4.5 M KCl) and stabilized with phalloidin. The actin network is spiked with ATP to 1 mM, and freshly thawed myosin is added. The actin-myosin mix is then centrifuged at 4 °C for 30 min at $128360 \times g$. The supernatant is collected, and optical absorbance is measured at 280 nm to determine the concentration.

### GUVs preparation
Actomyosin-encapsulated GUVs were prepared using the method described in previous studies[23,30,35,70]. All lipids were stored in chloroform at −20 °C. After preparing the lipids, the lipids were combined in a glass vial and dried under Argon gas. The lipids were dissolved in mineral oil (Sigma–Aldrich) at 2 mg/ml. The oil mixture was then sonicated in a bath sonicator for 2 h at room temperature. The mixture was then cooled to room temperature and stored at 4 °C. Then 7 μL of protein mix (actin polymerization buffer + proteins) was added to 70 μL of mineral oil in a 0.65 mL tube (07200185; Corning Costar). This mixture was then syringed in a glass syringe (Hamilton) 1–2 times. Separately, in a low absorption 0.65 mL tube, 30 μL of mineral oil was added to the top of 30 μL of OB. Then, the 60 μL emulsion was added to the top of the mineral oil layer in the tube. This mixture was then centrifuged at 100 g for 15 min at 4 °C. The 150 μL OB was injected to chambered coverslips. After the centrifugation, the upper oil layer was gently aspirated, and the bottom liposomes were aspirated. The liposomes were gently injected into OB. F-actin polymerization was allowed for ~10 min at R.T. (~25 °C) after preparing liposomes. Thereafter, myosin was light-activated, and the dynamics of membrane deformation were recorded by timelapse images with 6 s intervals. Liposomes were imaged by Leica DMi8 inverted microscope equipped with a 63× 1.4-NA oil immersion lens (Leica Microsystems).

It should be noted that there was no actin cortex formed outside the liposomes. Even if we assume all the liposomes were ruptured and proteins were dispersed in the OB, the outer protein concentrations will be less than 2% of the original concentration; thus, the outer cortex cannot be formed. To be further sure, OB does not contain KCl and with low $MgCl_2$ and $CaCl_2$ concentrations, making the capability of F-actin polymerization minimal. Additionally, we observed that the peak of the actin fluorescence intensity was located within the liposome membrane (Supplementary Fig. 6). Together, in our system, the possibility of F-actin polymerization outside of the liposome was negligible and would not contribute to the deformation.

Encapsulation efficiency of proteins within GUVs may vary with the encapsulation method, such as cDICE, with different studies employing specific protocols involving varied concentrations/contents, buffer compositions, and lipid compositions[71]. Although assessing encapsulation efficiency successfully is beyond the scope of this work, we have demonstrated rapid actomyosin contraction within liposomes with contraction timescales of minutes, comparable to two-dimensional assays[17,38], in a precisely controlled manner depending on myosin concentration. Thus, our system has established robust and efficient myosin incorporation without losing its activity.

### Estimation of the number of membrane binding sites for his-tagged proteins
Given the head group size of EPC to be 0.55 nm$^2$ [72], the total number of lipids in a lipid bilayer membrane of a GUV with a radius $R = 20$μm is $N_{tot} \sim 4\pi R^2/0.55 \sim 10^{10}$. Thus, the total number of DOGS-NiNTA lipids with fraction ~10% contained in the bilayer is $N_{tot}^{NiNTA} \sim 10^9$. On the other hand, the total number of 1 μM protein encapsulated within a GUV with $R = 20$μm is $N_{prot}^{1μM} \sim 10^7$. Thus, complete binding of the protein and DOGS-NiNTA lipids at a 1:1 ratio leaves $10^2$ times more available binding sites. However, the binding of the protein to the membrane may be limited by the size of the protein itself. For example, mDia1 with a molecular weight of 178 kDa has an approximate radius $R_{mDia1} \sim 4$nm, estimated based on molecular weight[73]. Thus, the maximum number of mDia1 that can be bound to the bilayer membrane is estimated to be $N_{mDia1}^{max} \sim 4\pi R^2/\pi R_{mDia1}^2 \sim 10^8$. Notably, this is still below the total number of 1 μM protein bound to the membrane estimated above ($N_{prot}^{1μM} \sim 10^7$). In

the case of VCA with a molecular weight of ~43 kDa and $R_{VCA} \sim 2$ nm, $N_{VCA}^{max} \sim 4\pi R^2 / \pi R_{VCA}^2 \sim 4 \times 10^8$. Thus, in our experimental conditions, all the his-tagged proteins can be bound to the membrane.

## Protein concentrations

The final concentration of non-fluorescent actin (AKL99-D; Cytoskeleton Inc.) rhodamine-actin (AR05-C; Cytoskeleton Inc.) and skeletal muscle myosin II (MY02; Cytoskeleton Inc.) were 5.3 μM, 0.7 μM, and 70–280 nM, respectively. In the experiments assembling branched and linear actin networks, Arp2/3 (RP01P-B; Cytoskeleton Inc.), His-mDia1 (Methods), His-VCA (Methods), and profilin (PR02; Cytoskeleton Inc.) were used at 25–75 nM, 25–625 nM, 1.5 and 4.5 μM, and 0.1–0.5 μM, respectively.

## Protein purification

His-tagged purified proteins were prepared by following the method described in previous studies[17,19]. Briefly, SNAP tagged mDia1(FH1C) containing a carboxy-terminal His(6x)-tag was expressed in *E.coli* (BL21-Codon Plus (DE3)-RP strain). The cells were induced with 500 μM IPTG at 16 °C and purified using Talon® metal affinity resin by following the manufacturer's instructions. The mammalian SNAP-mDia1 (FH1C)-HIS plasmid was a gift from David Kovar lab. N-WASP fragment WWA (aa400–501, also called VCA) containing a carboxy-terminal His(6x)-tag was expressed in *E.coli* (BL21-Codon Plus (DE3)-RP strain). The cells were induced with 1 mM IPTG for 2 h at 37 °C and purified using Ni-NTA resin by following the manufacturer's instructions, followed by another purification using the size exclusion chromatography. The pET17b KCK-ratWWA-6xHis plasmid was a gift from Cecile Sykes lab.

## Microscopy and data acquisition

Images were acquired by Leica DMi8 inverted microscope equipped with a 63× 1.4-NA oil immersion lens (Leica Microsystems), a spinning-disk confocal (CSU22; Yokogawa), and sCMOS camera (Zyla; Andor Technology) controlled by Andor iQ3 (Andor Technology). One pixel measured 0.0938 μm/pix. Images were recorded by timelapse with 6 s intervals for 2 min after focusing on the mid-plane of the liposomes at room temperature (~25°C). Exposure times were 800 ms for 405 nm laser (10% power, 10 mW) for inactivation of blebbistatin, 300 ms for 488 nm laser (8% power, 8 mW) for visualization of membrane, and 300 ms for 561 nm laser (10% power, 7.5 mW) for visualization of rhodamine-actin, respectively.

## Image analysis and data quantification

PIV analysis—Quantitative image analysis on contracting actomyosin network was performed by using a custom code written in MATLAB. We measure the displacement vectors between individual frames at every spatial location $u(r, t)$ using a public domain PIV program implemented as a Fiji/ImageJ plugin. The first, second, and third PIV window size and search window size were set to 100 and 200, 60 and 120, 48 and 96 pixels, respectively. The third PIV window size was 48 pixels for volume network contraction analysis, and 16 pixels or 32 pixels for cortical flow analysis. Correlation, noise, and threshold parameters were set to 0.6, 0.20, and 5, respectively. PIV was performed between each frame with 6 s intervals. Mean compressive strain $\varepsilon$ of the actin network is calculated from the divergence of the cumulative displacement field, $\varepsilon(t) = -\int_0^t dt' \int d\mathbf{r}[\nabla \cdot \mathbf{u}(\mathbf{r}, t')] / \int d\mathbf{r}$. The spatial integral is taken over the negative divergence within a droplet to calculate the compressive strain.

Nematic order parameter calculation - We measure the orientation field of F-actin using a public domain program OrientationJ implemented as a Fiji/ImageJ plugin. The image of the bottom surface of the liposomes was used to calculate the orientation field. The local nematic order parameter was computed from the director field around the reference point, $S = \langle \cos 2\theta \rangle$, where $\langle \cdot \rangle$ is the local average around the reference point. The four surrounding points were averaged to compute the local orientation order parameter. The mean nematic order parameter, $\langle S \rangle$, was calculated by averaging all the local nematic order parameters.

Contour analysis of the membrane deformation - The position of the membrane was extracted from the image of the membrane channel using a public domain program JFilament implemented as a Fiji/ImageJ plugin. The extracted contour position of the membrane was imported to MATLAB. The center-of-mass of the liposome, $\mathbf{r}_c = N^{-1}\Sigma\mathbf{r}_i$, is calculated from the position of the membrane, $\mathbf{r}_i = (x_i, y_i), i = 1 \cdots N$. The origin of the polar coordinate was then centered at the center-of-mass of the liposome by subtracting the center-of-mass of the contour position from each position of the membrane as $\mathbf{r}_i - \mathbf{r}_c$. The average radius $\langle R(\theta, t) \rangle_\theta$ of the liposomes was calculated by averaging the $R(\theta, t)$ for $-\pi < \theta < \pi$. The amplitude of the membrane deformation is defined as $u(\theta, t) = R(\theta, t) - \langle R(\theta, t) \rangle_\theta$. Power spectrum of the amplitude of the membrane deformation was computed by the squared Fourier transform of $u(\theta, t)$. Angular autocorrelation of $u(\theta, t)$, $\langle u(\theta + \Delta\theta, t)u(\theta, t) \rangle_\theta$, was calculated at each time point. Subsequently, the $\Delta\theta$ that satisfies $\langle u(\theta + \Delta\theta, t)u(\theta, t) \rangle_\theta = 0$ was extracted at each time point. Thereafter, the smallest value of $\Delta\theta$ at time point $t'$ is defined as the characteristic size of deformation, $\theta_c$. The local curvature of the membrane was determined by fitting a circle to the three separated points around each membrane position by the Pratt–Newton method. We analyzed the comparable sizes of liposomes with diameters ~35–55 μm to reduce the variability (Supplementary Fig. 2).

Cortex thickness analysis - We employed a well-established method used in cells for measuring cortex thickness[15,74]. Briefly, the cortex thickness is estimated through the distance between the position of the peak of the membrane fluorescence and that of the peak of the actin cortex intensity (Supplementary Fig. 6). Using the distance between the two peaks, $\Delta$, the cortex thickness, $h$, is ideally estimated as $h \simeq 2\Delta$. However, due to the presence of some actin fluorescence in the liposome volume, the position of the peak is slightly shifted toward the liposome center by a distance $\delta$[74]. The shift distance $\delta$ is given by $\delta = (\sigma^2/h)\ln[(I_{out} - I_S)/(I_V - I_S)]$, where $I_S$ is the actin cortex intensity, $I_V$ is the actin intensity in the liposome volume, $I_{out}$ is the intensity outside of the liposome, and the standard deviation (SD), $\sigma \simeq 119$ nm, of the point spread function of the microscope, was estimated by fitting a Gaussian function to the fluorescence of sub-resolution beads (Supplementary Fig. 7). Combining this with the relation, $h = 2(\Delta - \delta)$[74], the cortex thickness was estimated.

## Statistics and reproducibility

Statistical tests comparing distributions are done with the Wilcoxon rank sum test. All data displayed as a single value with an error bar is quoting the mean ± SD. Data are presented as boxplots where the interquartile range (IQR) is between Q1 (25th percentile) and Q3 (75th percentile), the center line indicates the median, whiskers are extended to Q3 + 1.5 × IQR and Q1 − 1.5 × IQR. The symbols *, **, and *** represent $p < 0.05$, 0.01, and 0.001 respectively. The sample size was determined by experimental feasibility and adjusted when it was deemed sufficient, ensuring that sample statistics were not affected by changes in sample size. Sample sizes (n) are indicated at the relevant locations in the manuscript. Experiments were independently replicated at least two times, and all replications were successful. Independent experiments (N) are indicated at the relevant locations in the manuscript.

## Reporting summary

Further information on research design is available in the Nature Portfolio Reporting Summary linked to this article.

## Data availability

Raw data supporting the findings of this manuscript are available from the corresponding authors upon request. The source data behind the graphs is available as Supplementary Data 1. A reporting summary for this Article is available as Supplementary Material.

## Code availability

Code supporting the findings of this manuscript is available from the corresponding authors upon request. Fiji (https://imagej.net/software/fiji/

downloads) was used for basic image processing. Particle image velocimetry (PIV) was performed in Fiji using a public domain plugin[75] (https://sites.google.com/site/qingzongtseng/piv#h.39ycjstb0wmn). MATLAB (R) version 2021b was used for data analysis and graph production. A reporting summary for this Article is available as Supplementary Material.

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

## Acknowledgements

The authors acknowledge funding ARO MURI W911NF-14-1-0403 and M.P.M, NIH RO1 GM126256 to M.P.M, NIH U54 CA209992 to M.P.M, and Human Frontier Science Program (HFSP) grant # RGY0073/2018 to M.P.M. M.P.M. and R.S. acknowledge support from Yale start-up funds. R.S. acknowledges support from the Overseas Postdoctoral Fellowships of the Uehara Memorial Foundation and Japan Society for the Promotion of Science (JSPS). Any opinions, findings, and conclusions or recommendations expressed in this material are those of the authors and do not necessarily reflect the views of the NSF, NIH, ARO, HFSP, Uehara Memorial Foundation, or JSPS.

## Author contributions

R.S. and M.P.M. designed the experimental work. R.S. acquired the experimental data. R.S. contributed new reagents/analytic tools. R.S. analyzed experimental data. R.S. drafted the paper. R.S. and M.P.M. edited the paper.

## Competing interests

The authors declare no competing interests.
