## [Peer Review file · Communications Biology]

Composite branched and linear F-actin maximize myosin-induced membrane shape changes in a biomimetic cell model

Corresponding Author: Professor Michael Murrell

Figures originally included in the author's rebuttal have been redacted from this file.

Version 0:

Reviewer comments:

Reviewer #1

(Remarks to the Author)

The study by Sakamoto and Murrell explores how the architecture of F-actin, either branched or linear, influences GUVs shape changes. The experiments indicate that neither branched nor linear F-actin alone can induce significant shape changes when myosin contractility is activated. However, mixing both actin nucleators leads to measurable membrane deformations.

Through quantitative analysis, the study highlights the importance of optimizing F-actin architecture for mimicking cellular behaviors. The authors discuss how their results compare to what is known in different cells, where a fine-tuned balance of the activity of Arp2/3 and formin is required during different stages of the cell cycle or during organism development.

The experiments seem to be carefully conducted and quantitatively analyzed. I understand that investigating the contractility of actomyosin networks in reconstituted liposomes allows for a clear readout of the actual force transmission by different F-actin networks, still I am having doubts about how much we are gaining in understanding compared to what was already known at the molecular scale.

Still, I am supportive for the work to be published, as long as the authors make some important clarification in the introduction and discussion.

Please find below my main observations and suggestions:

- 1 - For me there are two main features that allow actin filaments to actually transmit myosin contractility to the plasma membrane: first one is filament anchoring (density of anchoring points), second one is network connectivity. What is absolutely unclear to me is the way GUVs and protein mixing is done is how much VCA and how much formins are anchored, as well as how much formins are in solution. In the paper, the authors do discuss about the fact that part of the formins are not membrane-bound and actively contribute to filament nucleation in bulk, but that is never quantitatively assessed. Thus, this makes the assessment of the results for different ratios of mDia1/Arp23 quite difficult indeed.
- 2 - I understand that varying the amount of Arp2/3 is a way to tune the branching network, but why not do it with VCA instead?
- 3 - What is not clearly mentioned is whether there is VCA in the absence of Arp2/3? This matters, as VCA can compete with formins for barbed ends, through the barbed end interference mechanism recently and very elegantly highlighted by the Bieling lab (Funk et al, Nat. Comm., 2021). Can the authors clarify this point?
- 4 - In the discussion, the authors could also mention/discuss what is happening in the sole presence of mDia1: do formins unbind from barbed ends as myosin is pulling on filaments, or are formins pulled together with the filament towards the myosin-induced asters?
- 5 - One important missing point is a clear discussion on the actual architecture of mixed networks to get an effective liposome deformation: is this because long linear filaments get branched by VCA-Arp2/3? Interestingly, to have such a clear cut effect on myosin contractility to be effectively transmitting force by a mixed network, this would mean that the length of filaments elongated by mDia1 is order(s) of magnitude longer than that of filaments with free elongating barbed ends, and some percolation happens. This is somehow reminiscent of what has been seen in Schuppler et al, 2016, figure 5. Do the

have an estimate in the change of filament length in the various conditions tested ? similarly an estimate of the distance between branching points would be also interesting.

6 - Can the authors discuss what explain non-slip boundary conditions for the Arp2/3 only networks ? Is it a matter of filament attachment point density ? As shown in fig5a, having too much Arp2/3 (diamonds) compared to a limited amount of Arp2/3 (25nM, colored dots) seem to be key to get membrane strain or not.

7 - How these results would be affected by the presence of a crosslinker, such as alpha-actinin, should be discussed.

minor points:

- What is the reason to not have included CP in the assay ?

- supp.movie legends should indicate the protein concentration, at least for Arp2/3 and mDia1.

- Can the authors clearly mention the amount of VCA or mDia1 that are expected to be bound to NTA lipids ?

- line 228: "[Arp2/3] = 25 nM; [mDia1] = 25 nM)" which is different from what is mentioned in the methods (μM range).

- move section 'Actin cortex architecture determines force generation, transmission, and membrane deformations.' (line 287) to discussion part.

- line 302: 'there is no crosslink between between the F-actin and mDia1' : what do you mean here ?

- line 267: grammar issue.

- line 278 : I would not derive the same conclusion as the authors about the effect of profilin on formin filament cunleation : profilin decreases the efficiency of mDia1 nucleation, while accelerating barbed end elongation rate. So the net effect is not that straightforward, and can very well be a bell shape curve for the total amount of F-actin being assembled in the presence of profilin.

- line 337 : ref. 57 seems not correct. Did the authors meant ref. 15 ? the same remark applies two lines after.

- line 328: "When the branched F-actin network is mixed with the linear F-actin network" : is a rather 'clumsy' sentence. To be replaced by 'composite network' as used two lines after.

Reviewer #2

(Remarks to the Author)

This manuscript by Sakamoto and Murrell investigates the influence of linear, branched and mixed actin architectures on the membrane deformation of GUVs and actin polarity. To do so, the authors develop an elegant system to induce the contraction of actomyosin networks upon light illumination inside vesicles using blebbistatin inactivation. This offers full spatio-temporal control and allows them to investigate different actin architectures. They show that only mixed linear and branched actin networks induce membrane deformation of GUVs. The authors interpret these findings to be caused by different no-slip (branched) or slip (linear) boundary conditions of the networks enabling different degrees of force transmission and generation. Overall, the results are well-balanced between experimental insights and their careful analysis including the approximation of several important biophysical quantities. The conclusions from the data are generally justified and make this work interesting for the protein reconstitution and potentially cell biology community. However, I have several comments that need to be addressed before I can recommend its publication.

Comment 1: The authors use blebbistatin inactivation using laser light to initiate the actomyosin contraction. I would encourage the authors to perform to additional control experiments to verify that the contraction in their system is indeed occurring due to the proposed mechanism: i) with laser activation but in absence of blebbistatin and ii) with laser activation and blebbistatin but in absence of myosin.

Comment 2: In Figure 2 the authors show that only the mixed actin cortex can induce deformation of GUVs. I worry that this effect is caused by an increase of membrane linkers and not the mixing of linear and branched actin. The authors add 25nM of each His-VCA and His-mDia1 and thus twice as many membrane anchoring sites compared to the experiments with only linear or branched networks. I think the authors should compare the effect of the different actin architectures at a constant density of membrane anchors.

Comment 3: In the methods section the authors mention that there is an osmolarity difference between the inner and outer buffers but do not specify if the vesicles are deflated or inflated. I would ask them to add this information and explain why this was chosen instead of iso-osmotic conditions.

Comment 4: The authors observe membrane deformations in the presence of mixed actin networks. Where does the excess membrane go? Do the authors observe lipid nanotubes forming during the contraction?

Comment 5: Do the membrane-bound actin networks have different dynamics? The authors could test this by photobleaching the respective actin networks and compare their recovery traces. I suspect the Arp2/3 networks to be much more static if the no-slip condition is valid.

Comment 6: The inverted emulsion method to form GUVs has a relatively low yield. Can the authors exclude that a cortex forms also on the outside of the GUVs which contributes to the mechanical deformation?

Comment 7: I very much appreciate that the authors include an estimate of the cortex tension. However, I do not think that the estimate of the cortex thickness h in Supporting Figure 3b is reliable because it is on the order of the resolution limit of the microscope. Thus, the cortex thickness could be well below 500nm. Since the bending rigidity, stretching modulus and cortex tension scale with h , I am afraid that they are overestimated. Could the authors further justify or provide additional

evidence of their estimate of the cortex thickness?

Comment 8: The authors use NTA-tagged lipids to anchor the cortex to the membrane. Could the authors add an estimate of how many NTA-lipids are bound to His-tagged VCA or mDia1 assuming complete binding? How many more binding sites are there compared to His-tagged molecules?

Minor comments:

In my opinion the title is too far-fetched because the authors study GUVs and no biomimetic model cells. Thus, I suggest to change the title accordingly.

Typo I. 65 unilamellar

Typo I.165 These

Reviewer #3

(Remarks to the Author)

The manuscript 'Composite branched and linear F-actin maximize myosin-induced membrane shape changes in biomimetic model cells' by Sakamoto and Murrell describes membrane shape deformations of liposomes induced by different types of actin-myosin networks that are reconstituted within. The study shows very nicely that a mixture of branched (Arp2/3) and straight (mDia1) actin filaments provide the best substrate for myosin II motors to induce large scale membrane deformations that can be easily detected using confocal fluorescence microscopy. Quantitative image analysis is elegantly used to determine the cortical strain and compute forces and energies that lead to the observed deformations.

This work is a nice continuation of the pioneering work on reconstituted actin-myosin networks of the Murrell lab and provides important novel insights for the community how to realize reconstituted actin networks that enable GUV deformation.

Major points:

From the movies it seems that contractions seem to concentrate on one or two locations, conveniently at the equatorial plane. Were GUVs observed/ tested for contractions elsewhere? Or is the blebbistatin inactivation limited to the equatorial plane?

The authors limit their analysis to GUVs larger than 20 microns. Could they comment on the fate of smaller GUVs? Are they more likely to simply collapse upon myosin activation?

Reproducibility: some measurements, e.g. Fig 3e, f, h; Fig 4 c-h; sup fig 1a; sup fig 3 c-e; sup fig 4 a-e, are based on one-time experiments (N=1) without an independent repeat. I am happy to believe that these results are reproducible, however, I think that results shown in a publication should be based on multiple (at least two) independent experiments.

Polarized actin flow: The authors mention having observed polarized actin flow in 12 % of liposomes (sup fig. 2e). To make this comparison, one would assume to assess the ratio of GUVs with actin flow compared to the whole number of GUVs that were tested in each experiment. But the figure legend mentions N = 2 for the case of non-polarized GUVs and N=4 for polarized. Or is this a confusion due to similar terms for different things, i.e. the authors compare GUVs prepared with a non-polarized (Arp2/3) and a polarized (Arp2/3 + mDia1) actin network, and ask then, how many of these GUVs show polarized flow?

It is mentioned in the methods that the buffer within the GUVs contains 5 mM ATP and an ATP regeneration system. The actin network shows immediate network contraction upon light activation. However, myosin activity seems to cease within the first three minutes. Given the high ATP concentration, one could imagine that myosin activity would be sustained for longer periods. Can it be that the long time of 405 nm light illumination also leads to photodamage of myosin? Can the authors comment on the levels of ATP within the GUV after actin polymerization?

In addition to that, can the authors comment on the efficacy of myosin incorporation with their approach compared to other encapsulation techniques?

Minor points:

Page 10, line 295: write 0.5 instead of $\frac{1}{2}$ to be consistent with the text above.

Refs 35 and 49 are the same!

Fig 5a: the phase diagram might be easier to read when adding a third dimension that describes [mDia1]/[Arp2/3] ratio.

PIV: please provide the parameters used for PIV analysis, i.e. interrogation box size (single or multiple) and step size.

Sup Fig 2: a-c: it is a bit surprising to see the width of actin flow in the PIV analysis considering the actin is supposed to move along the membrane. Could the authors comment on the reason for this? Are the parameters set too sensitive? Is this due to the overall membrane deformation that occurs within the first minute?

f-h: it seems colors for polarized and non-polarized were switched between panels.

Author Rebuttal letter:

Response Letter

Composite branched and linear F-actin maximize myosin-induced membrane shape changes in a biomimetic cell model

We greatly appreciate all the reviewers for carefully reading our manuscript and for their insightful comments. In light of their suggestions, we have made changes to the manuscript and responded to each comment below.

Response to Reviewer #1

The study by Sakamoto and Murrell explores how the architecture of F-actin, either branched or linear, influences GUVs shape changes. The experiments indicate that neither branched nor linear F-actin alone can induce significant shape changes when myosin contractility is activated. However, mixing both actin nucleators leads to measurable membrane deformations.

Through quantitative analysis, the study highlights the importance of optimizing F-actin architecture for mimicking cellular behaviors. The authors discuss how their results compare to what is known in different cells, where a fine-tuned balance of the activity of Arp2/3 and formin is required during different stages of the cell cycle or during organism development. The experiments seem to be carefully conducted and quantitatively analyzed. I understand that investigating the contractility of actomyosin networks in reconstituted liposomes allows for a clear readout of the actual force transmission by different F-actin networks, still I am having doubts about how much we are gaining in understanding compared to what was already known at the molecular scale.

Still, I am supportive for the work to be published, as long as the authors make some important clarification in the introduction and discussion.

We are grateful to the reviewer for identifying our work carefully conducted and quantitatively analyzed. We agree that some experimental conditions and discussions were not clarified enough in the initial manuscript. We first added a separate paragraph in the Introduction section to clarify the motivation of this study (pp. 3, lines 55-63). According to your suggestions, we made the following revisions:

Please find below my main observations and suggestions:

1. For me there are two main features that allow actin filaments to actually transmit myosin contractility to the plasma membrane: first one is filament anchoring (density of anchoring points), second one is network connectivity. What is absolutely unclear to me is the way GUVs and protein mixing is done is how much VCA and how much formins are anchored, as well as how much formins are in solution. In the paper, the authors do discuss about the fact that part of the formins are not membrane-bound and actively contribute to filament nucleation in bulk, but that is never quantitatively assessed. Thus, this make the assessment of the results for different ratios of mDia1/Arp23 quite difficult indeed.

We appreciate this comment. We acknowledge the way we mix GUV and proteins were not sufficiently clear in the initial manuscript and lead to some confusion. First, in the methods

1 section we described the way we prepare GUV, in which we encapsulate the proteins inside the GUVs using the inverted emulsion methods. To avoid confusion, we separately prepared the Methods section \hat{a} GUV preparation \hat{a} and detailed the encapsulation procedure (pp. 15, lines 473-503). It should be noted that both mDia1 and VCA are his-tagged and the lipid bilayer membrane contains DGS-NiNTA lipids. Therefore, all His-mDia1 and His-VCA are assumed to be anchored to the lipid bilayer membrane, as established in our previous studies (Vikrant et al., Adv. Funct. Mater. 29, 1905243 (2019); Muresan et al., Nat Commun. 13, 7008 (2022); Sakamoto et al., Commun. Biol. 6, 325 (2023)). We have clarified this experimental design in the Results section to avoid confusion (pp. 4, lines 100-102).

Additionally, we have estimated the number of membrane binding sites available for His-tagged protein under our experimental conditions, suggesting that there are sufficient binding sites for all His-tagged proteins. Given the head group size of EPC as 0.55 nm² (Lasic, CRC Press (1997)), the total number of lipids in a lipid bilayer membrane of a GUV with a radius $\delta = 20 \text{ \AA}$ is $\delta^2 / 0.55 \hat{a} 1010$. Thus, the total number of DOGS-NiNTA lipids at a NiNTA

$\sim 10\%$ fraction in the bilayer is $\delta^2 \hat{a} 109$. On the other hand, the total number of 1 \AA M

$1 \hat{a} 4 \text{ M}$ protein encapsulated within a GUV with $\delta = 20 \text{ \AA}$ is $\delta^2 \hat{a} 107$. Therefore, complete binding of the protein and DOGS-NiNTA lipids at a 1:1 ratio leaves 102 times more available binding sites. However, protein binding may be limited by the protein size. For instance, mDia1 (178 kDa) has an estimated radius of $\delta_{\text{mDia1}} \hat{a} 4 \text{ nm}$ based on its molecular weight (Erickson, H. P., Biol. Proced. Online 11, 32-51 (2009)). Thus, the maximum number of mDia1 that can

max 2 bind to the membrane is $\delta_{\text{mDia1}}^2 \hat{a} 4 \delta^2 / \delta_{\text{mDia1}} \hat{a} 108$. Notably, this is still below the total

$1 \hat{a} 4 \text{ M}$ number of 1 \AA M protein bound to the membrane estimated above ($\delta^2 \hat{a} 107$). In the case

max 2

of VCA with a molecular weight of ~43 kDa and δ VCA \approx 2 nm, δ VCA \approx 4 δ / δ VCA \approx 4 Å 108 . This indicates that all His-tagged proteins can bind to the membrane under our conditions. We have added this estimation in the Methods section to clarify the experimental condition (pp. 16, lines 504-517) and mentioned in the Results section (pp. 4, lines 102-104). Also, the reviewer mentioned "the formins are not membrane-bound and contribute to filament nucleation in bulk"; however, we did not intend to mean it that way, and we agree that the expression in the previous manuscript was misleading. What we meant is that the nucleated F-actin can be released from "membrane-bound His-mDia1" and the released F-actin may exist in the liposome volume, as previously observed (Vikrant et al., *Adv. Funct. Mater.* 29, 1905243 (2019)). To clarify this point, we have revised the sentence in the Results section to avoid potential confusion (pp. 5, lines 128-129; pp. 10, 279-281).

2. I understand that varying the amount of Arp2/3 is a way to tune the branching network, but why not do it with VCA instead ?

Thank you for this comment. The method of varying Arp2/3 concentration to tune the architecture of branched F-actin network was characterized and optimized in our previous

2 study (Muresan et al., *Nat Commun.* 13, 7008 (2022)). Therefore, we employed this method. It should be noted that we have tested a higher His-VCA concentration in the results section, which exhibited membrane strain comparable to the control condition, indicating that stronger actin-membrane links may facilitate force transmission and membrane deformation (Supplementary Fig. 10) (pp. 10, lines 302-305). A detailed investigation of the influence of different VCA concentration on F-actin architecture and membrane deformation capability would be a valuable future direction.

3. What is not clearly mentioned is whether there is VCA in the absence of Arp2/3 ? This matters, as VCA can compete with formins for barbed ends, through the barbed end interference mechanism recently and very elegantly highlighted by the Bieling lab (Funk et al, *Nat. Comm.*, 2021). Can the authors clarify this point ?

We appreciate this insightful comment. There was no His-VCA present in the absence of Arp2/3. We have added this point in the Results section to clarify the experimental setup (pp. 5, lines 126). Also, by following the reviewer's suggestion, we have mentioned the potential competition mechanism between VCA and formins in the Discussion section with citing the suggested paper (Funk et al., *Nat. Commun.* 12, 5329 (2021)) (pp. 13, lines 399-400).

4. In the discussion, the authors could also mention/discuss what is happening in the sole presence of mDia1 : do formin unbind from barbed ends as myosin are pulling on filaments, or are formins pulled together with the filament towards the myosin-induced asters ?

We appreciate this insightful comment. It has been shown in two-dimensional system that the filaments nucleated by mDia1 can be well-contracted and form asters (Muresan et al., *Nat Commun.* 13, 7008 (2022)). However, it is challenging to clarify if mDia1 is unbound or bound together with myosin-induced asters in our experimental setup, which is also beyond the scope of this work. As the reviewer suggested, it would be possible that mDia1 may unbind from barbed ends or mDia1 pulled together with the filament towards the myosin-induced asters. Contracting polar-aster-like structures may be generated by the cooperation of myosin contraction and formin-based actin nucleation as observed in cells (Costache et al., *Cell Reports* 39, 110868 (2022)). Additionally, the force-sensing property of mDia1 may play a role in the contractile behavior during aster formation (JÃ©gou et al., *Nat. Commun.* 4, 1883 (2013); Yu et al., *Nat. Commun.* 8, 1650 (2017)). We have added these points in the Discussion section to clarify the possible ways of network contraction (pp. 12, lines 380-386).

5. One important missing point is a clear discussion on the actual architecture of mixed networks to get an effective liposome deformation: is this because long linear filaments get branched by VCA-Arp2/3 ? Interestingly, to have such a clear cut effect on myosin contractility to be effectively transmitting force by a mixed network, this would mean that the length of

3 filaments elongated by mDia1 is order(s) of magnitude longer than that of filaments with free elongating barbed ends, and some percolation happens. This is somehow reminiscent of what has been seen in Schuppler et al, 2016, figure 5. Do they have an estimate in the change of filament length in the various conditions tested ? similarly an estimate of the distance between branching points would be also interesting.

Thank you for this insightful comment. As the reviewer suggested, we believe that the mixed architecture network may facilitate force transmission due to the linear network branched via Arp2/3 and force percolation effectively occurs, as previously observed

(Schuppler et al., Nat. Commun. 7, 13120 (2016); Alvarado et al., Nat. Phys. 9, 591-597 (2013)). Although it is challenging to estimate filament length within liposomes, single molecule experiments and theoretical modeling studies have suggested that mDia1-nucleated filament length is 10 times longer than Arp2/3-nucleated filament length in the HeLa cell cortex, and the filament length was found to be insensitive to the fraction of Arp2/3 and mDia1 (Fritzsche et al., Sci. Adv. 2, e1501337 (2016)). Thus, in the mixed F-actin architecture in our study, the length of F-actin elongated by mDia1 could be an order of magnitude longer than that of F-actin with free elongating barbed ends, potentially leading to some percolations. A detailed characterization of filament length distribution within liposomes, together with theoretical modeling, would be an important future challenge. We have added these points in the Discussion section to clarify the potential architectural mechanism of membrane deformation in the mixed architecture liposome (pp. 13, lines 391-397).

6. Can the authors discuss what explain non-slip boundary conditions for the Arp2/3 only networks ? Is it a matter of filament attachment point density ? As shown in fig5a, having too much Arp2/3 (diamonds) compared to a limited amount of Arp2/3 (25nM, colored dots) seem to be key to get membrane strain or not.

We appreciate this suggestion. There are multiple possible ways that the Arp2/3 network prevents actomyosin contraction, thereby functioning as an effective no-slip boundary condition, as referred to in this study. The Arp2/3-only networks could inhibit myosin force generation, as shown in our previous study, by limiting myosin motion through dense branching (Muresan et al., Nat Commun. 13, 7008 (2022)). Additionally, the viscoelasticity of the branched actin network and hinder clustering and resulting deformation (Pujol et al., PNAS 109, 10364-10369 (2012); Muresan et al., Nat Commun. 13, 7008 (2022)). We have added this point in the Discussion section to clarify the possible origin of the non-slip boundary-like behavior in the Arp2/3 only network (pp. 12, lines 366-368; pp. 12, lines 374-375).

Minor points:

7. What is the reason to not have included CP in the assay ?

Our group developed and optimized the F-actin polymerization assays using membrane-

bound mDia1 and Arp2/3 in the absence of CP (Vikrant et al., Adv. Funct. Mater. 29, 1905243 (2019); Muresan et al., Nat Commun. 13, 7008 (2022); Sakamoto et al., Commun. Biol. 6, 325 (2023)). Therefore, we did not include CP such as CapZ in our assay. Since CP has been reported to induce local membrane deformation in liposomes (D'Arre et al., Nat. Commun. 9, 1630 (2018)), the investigation of the impact of CP in combination with different nucleators would be an interesting future direction. We have mentioned the possibility of the investigation using CP and cited this article in the Discussion section to indicate the possible future research (pp. 14, lines 428).

8. supp.movie legends should indicate the protein concentration, at least for Arp2/3 and mDia1.

We have added the concentrations of Arp2/3 and mDia1 in the movie legends accordingly.

9. Can the authors clearly mention the amount of VCA or mDia1 that are expected to be bound to NTA lipids ?

As described in Comments #1, we have estimated the number of membrane binding sites available for His-tagged protein under our experimental conditions, suggesting that that all the His-VCA and His-mDia1 is membrane bound. We have clarified this point in the results section (pp. 4, lines 100-104).

10. line 228: "([Arp2/3] = 25 nM; [mDia1] = 25 nM)" which is different from what is mentioned in the methods (ÅμM range).

It was a typo in the Methods section, which was corrected to nM range accordingly.

11. Move section 'Actin cortex architecture determines force generation, transmission, and membrane deformations.' (line 287) to discussion part.

We appreciate the suggestion. However, we believe that this section should be in the Results section to conceptualize the findings in our study. Thus, we would like to retain this section in the Results section.

12. line 302: 'there is no crosslink between the F-actin and mDia1' : what do you mean here ?

We meant the architectural difference between the Arp2/3 network and mDia1 network, while we acknowledge that the sentence was not clear. We revised this sentence to "since there is no branching or crosslinking within the mDia1-nucleated linear F-actin network, unlike the Arp2/3 cortex, the adjacent membrane acts ...â" in the Results section to clarify its meaning (pp. 11, lines 344-345).

13. line 267: grammar issue.

5

The grammar was corrected accordingly (pp. 10, lines 302).

14. line 278 : I would not derive the same conclusion as the authors about the effect of profilin on formin filament nucleation : profilin decreases the efficiency of mDia1 nucleation, while accelerating barbed end elongation rate. So the net effect is not that straightforward, and can very well be a bell shape curve for the total amount of F-actin being assembled in the presence of profilin.

We appreciate this insightful comment. We acknowledge that the initial manuscript did not provide enough discussion on the influence of profilin on F-actin nucleation. It has been shown that there is a biphasic bell-shaped dependence of the barbed end elongation rate on profilin concentration (Kovar et al., Cell 124, 423-435 (2006); Neidt et al., J. Biol. Chem. 284, 673-684 (2009)). This biphasic dependence has been attributed to the competing effects of accelerating the barbed-end elongation rate while excess profilin competes with profilin-actin for binding to formin, thus inhibiting nucleation (Vavylonis et al., Mol. Cell 21, 455-466 (2006); Kovar et al., Cell 124, 423-435 (2006)). Notably, for mDia1, the elongation rate increases with profilin concentration in the regime of profilin to actin concentration ratio $[Profilin]/[Actin] < 2.0$ (Kovar et al., Cell 124, 423-435 (2006)). Since we used a profilin to actin ratio at $[Profilin]/[Actin] < 0.1$, the observed bulkier F-actin distribution at higher profilin concentration is presumably attributed to the enhanced elongation rate induced by higher profilin concentration. We have included this extensive discussion in the Results section to clarify the possible effect of profilin concentration on formin filament nucleation (pp. 10-11, lines 312-319).

15. line 337 : ref. 57 seems not correct. Did the authors meant ref. 15 ? the same remark applies two lines after.

The automatic generation of citation made this typo. We have corrected the references accordingly.

16. line 328: "When the branched F-actin network is mixed with the linear F-actin network" : is a rather 'clumsy' sentence. To be replaced by 'composite network' as used two lines after. We have replaced the sentence as "In the composite branched and linear F-actin network," (pp. 11, lines 346).

6

Response to Reviewer #2

This manuscript by Sakamoto and Murrell investigates the influence of linear, branched and mixed actin architectures on the membrane deformation of GUVs and actin polarity. To do so, the authors develop an elegant system to induce the contraction of actomyosin networks upon light illumination inside vesicles using blebbistatin inactivation. This offers full spatio-temporal control and allows them to investigate different actin architectures. They show that only mixed linear and branched actin networks induce membrane deformation of GUVs. The authors interpret these findings to be caused by different no-slip (branched) or slip (linear) boundary conditions of the networks enabling different degrees of force transmission and generation. Overall, the results are well-balanced between experimental insights and their careful analysis including the approximation of several important biophysical quantities. The conclusions from the data are generally justified and make this work interesting for the protein reconstitution and potentially cell biology community. However, I have several comments that need to be addressed before I can recommend its publication.

We are grateful to the reviewer for identifying our work as an elegant system and could be interesting for the protein reconstitution and potentially cell biology community. We made the following revisions according to the reviewer's comments:

Main comments:

1. The authors use blebbistatin inactivation using laser light to initiate the actomyosin contraction. I would encourage the authors to perform additional control experiments to verify that the contraction in their system is indeed occurring due to the proposed mechanism: i) with laser activation but in absence of blebbistatin and ii) with laser activation and blebbistatin but in absence of myosin.

We appreciate the reviewer's suggestion. Following your suggestion, we have conducted additional experiments (ii) in the absence of myosin ($[Myosin] = 0 \text{ nM}$), as shown in Revised Figure 1e-h. The new experimental data confirm that myosin is required for light-activated contraction within liposome. Additionally, we performed experiments (i) in which myosin had already contracted during the preparation of liposomes (~30 min) in the absence of blebbistatin, as shown in Supplementary Fig. 1. Therefore, light activation is not necessary for myosin to contract when blebbistatin is absent. Indeed, overcoming this behavior was the initial motivation for our study: to inactivate myosin contraction during liposome preparation. Together, these additional control data strengthen the validity and robustness of the present experimental setup.

7
 10⁻²
 e 1
 [Myosin] (nM)
 f 1.5
 [Myosin] (nM)
 280 280

(s⁻¹)
 0.8 140
 140
 70 70
 1
 0.6 0 0

Strain,

Strain rate,
 0.4
 0.5
 0.2

0 0
 0 30 60 90 120 0 30 60 90 120
 Time (s)
 Time (s) Time (s)
 Time (s)

g 2
 h
 *** ** * 0.025 *** ** *

Max. strain rate (s⁻¹)
 1.5 0.02
 Max. strain

0.015
1
0.01
0.5
0.005

0 0
0 70 140 280 0 70 140 280
[Myosin] (nM) [Myosin] (nM)

8

2. In Figure 2 the authors show that only the mixed actin cortex can induce deformation of GUVs. I worry that this effect is caused by an increase of membrane linkers and not the mixing of linear and branched actin. The authors add 25nM of each His-VCA and His-mDia1 and thus twice as many membrane anchoring sites compared to the experiments with only linear or branched networks. I think the authors should compare the effect of the different actin architectures at a constant density of membrane anchors.

We appreciate this comment. The reviewer mentioned that doubling the number of linkers could potentially induce deformation. However, we ruled out this possibility in Fig. 4 by increasing the His-mDia1 concentration from 5 to 25 times in the mDia1 cortex. Even at 25 times higher His-mDia1 concentration, membrane deformation was not observed for mDia1-only liposomes. Thus, it is clear that deformation is not caused by the increased anchoring sites due to His-mDia1 but rather by the mixed architecture effect. Additionally, if we decrease the Arp2/3 concentration, the impact of the branched F-actin architecture will be reduced. Therefore, for the purpose of studying the effects of F-actin architecture, we retained the Arp2/3 concentration the same as the Arp2/3-only network in Fig. 2. We have added these explanation in the Results section to clarify the intention behind the experimental condition (pp. 5, lines 138-139; pp. 10, lines 289-293).

3. In the methods section the authors mention that there is an osmolarity difference between the inner and outer buffers but do not specify if the vesicles are deflated or inflated. I would ask them to add this information and explain why this was chosen instead of iso-osmotic conditions.

Thank you for this comment. Osmolarity of the outer buffer was chosen to be slightly higher than the inner buffer to make the liposome slightly deflated, which enables membrane deformation such as through wrinkling. We have clarified this point in the Methods section (pp. 15, lines 459-461).

4. The authors observe membrane deformations in the presence of mixed actin networks. Where does the excess membrane go? Do the authors observe lipid nanotubes forming during the contraction?

We appreciate this comment. In our understanding, the membrane deformation does not lead to excess membrane, but rather, the excess membrane presumably allows the membrane wrinkling-like deformation not nanotubes, as observed in Fig. 2. We chose a slightly higher osmolarity of the outer buffer to make liposomes slightly deflated to allow membrane deformation/wrinkling (Methods). We have added this point in the Results section to clarify the possible origin of membrane deformation (pp. 6, lines 143-144).

5. Do the membrane-bound actin networks have different dynamics? The authors could test

9
this by photobleaching the respective actin networks and compare their recovery traces. I suspect the Arp2/3 networks to be much more static if the no-slip condition is valid.

We appreciate this comment. We acknowledge that the assessment of actin dynamics would be interesting. Unfortunately, it is an extensive project since our aim in this is to study the impacts of F-actin architecture on the transmission and generation of myosin-induced active stress against membrane deformation, but not the turnover of actin network itself. Given that actin turnover rates are linked to the viscoelastic properties of the actin cortex (Salbreux et al., Trends Cell Biol. 22, 536-545 (2012)), exploring how varying turnover rates influence

contraction dynamics and membrane deformation such as blebbing, would also be an intriguing avenue for future research. We added this point in the Discussion section (pp. 12, lines 374-378).

6. The inverted emulsion method to form GUVs has a relatively low yield. Can the authors exclude that a cortex forms also on the outside of the GUVs which contributes to the mechanical deformation?

Thank you for this comment. It should be noted that the inverted emulsion method does not have proteins outside, so that we have never observed the actin cortex formed outside. Additionally, we observed that the peak of the actin fluorescence intensity was located within the liposome membrane (Supplementary Fig. 6). Moreover, even if we assume all the liposomes were ruptured and proteins were dispersed in the outer buffer, the outer protein concentrations will be less than 2% of the original concentration; thus, the outer cortex cannot be formed. To be further sure, outer buffer does not contain KCl and with low MgCl₂ and CaCl₂ concentration, making the capability of F-actin polymerization minimal. Together, in our system, the possibility of F-actin polymerization outside of the liposome was negligible and will not contribute to the deformation. We have added this point in the Methods section to clarify the experimental design (pp. 16, lines 489-496).

7. I very much appreciate that the authors include an estimate of the cortex tension. However, I do not think that the estimate of the cortex thickness h in Supporting Figure 3b is reliable because it is on the order of the resolution limit of the microscope. Thus, the cortex thickness could be well below 500nm. Since the bending rigidity, stretching modulus and cortex tension scale with h , I am afraid that they are overestimated. Could the authors further justify or provide additional evidence of their estimate of the cortex thickness?

We appreciate this insightful comment. In the initial manuscript, we analyzed the cortex thickness based on the full-width at half maximum (FWHM) of the fluorescence intensity. We acknowledge that this was a crude estimate, especially for objects smaller than the diffraction limit of the microscope (~300 nm). Therefore, we employed an alternative method for measuring cortex thickness, which has been well-established for measuring cell cortex thickness (Clark et al., Biophys. J. 105, 570-580 (2013); Chugh et al., Nat. Cell Biol. 19, 689-697 (2017)). Briefly, the cortex thickness is estimated through the distance between the position of the peak of the membrane fluorescence and that of the peak of the actin cortex intensity (Supplementary Fig. 6c). Using the distance between the two peaks, \hat{l} , the cortex thickness, \hat{a} , is ideally estimated as $\hat{a} = 2\hat{l}$. However, due to the presence of some actin fluorescence in the liposome volume, the position of the peak is slightly shifted toward the liposome center by a distance δ_z (Clark et al., Biophys. J. 105, 570-580 (2013)). The shift distance δ_z is given by $\delta_z = (\delta / \hat{a}) \ln[(\delta_{out} \hat{a} \delta) / (\delta \hat{a} \delta_{out})]$, where δ is the actin cortex intensity, δ_{out} is the actin intensity in the liposome volume, δ_{out} is the intensity outside of the liposome, and the standard deviation, $\delta = 119$ nm, of the point spread function of the microscope is estimated by fitting a Gaussian function to the fluorescence of sub-resolution beads (Supplementary Fig. 7). Combining this with the relation, $\hat{a} = 2(\hat{l} - \delta_z)$ (Clark et al., Biophys. J. 105, 570-580 (2013)), the cortex thickness can be estimated more accurately compared to the previous FWHM measurement. The updated analysis yielded a cortex thickness of $\hat{a} = 0.29 \pm 0.09 \text{ }\mu\text{m}$ at [mDia1]:[Arp2/3]=1:1, which was smaller than the initial estimates of $\hat{a} \approx 0.46 \text{ }\mu\text{m}$. We have updated the cortex tension estimates using the revised value in the main text (pp. 8, lines 238-245, pp. 8, lines 253-256). Since the order of magnitude of \hat{a} is the same as the initial estimate, this change does not alter the conclusion of the manuscript. We have re-analyzed all the cortex thickness analysis data using this method and included in Supplementary Fig. 6 and Supplementary Fig. 9. This analysis method was described in the Methods section (pp. 18, lines 577-586).

a b c
Actin 200

150 Actin cortex
intensity Actin Position ($\hat{A}\mu\text{m}$)
Membrane 4.4 3.8 4 4.2
100 0
Actin Membrane
1 1
0.2
Bulk (volume)
0.8 0.4
50 actin intensity
0.6 0.6

0.5
 0.4 0.8
 Actin
 0 Cortex thickness 0.2 1
 Membrane
 0 2 4 6 8 0 0
 3.4 3.6 3.8 4 4.2 4.4
 Position ($\text{\AA}\mu\text{m}$)
 Position ($\text{\AA}\mu\text{m}$)

d 1
 e 2
 f 1
 n.s. n.s. n.s.
 n.s. n.s. *** *** *** n.s.
 n.s. n.s. n.s. n.s.
 **
 Cortex thickness ($\text{\AA}\mu\text{m}$)
 Cortex thickness ($\text{\AA}\mu\text{m}$)
 Cortex thickness ($\text{\AA}\mu\text{m}$)

n.s. n.s.
 0.8 n.s.
 * 0.8
 1.5

0.6 0.6
 1
 0.4 0.4

0.5
 0.2 0.2

0 0 0
 [Arp2/3] 25 0 25 0 25 0 (nM) [Arp2/3] 25 75 25 75 25 75 (nM)
 [mDia1] 25 25 125 125 625 625 [mDia1] 0 0 25 25 125 125
 [mDia1]/[Arp2/3]

8. The authors use NTA-tagged lipids to anchor the cortex to the membrane. Could the authors add an estimate of how many NTA-lipids are bound to His-tagged VCA or mDia1 assuming complete binding? How many more binding sites are there compared to His-tagged molecules?

Thank you for this insightful comment. Following your suggestion, we have estimated the number of membrane binding sites available for His-tagged protein under our experimental conditions, suggesting that there are sufficient binding sites for all His-tagged proteins. Given the head group size of EPC to be 0.55 nm^2 (Lasic, CRC Press (1997)), the total number of lipids in a lipid bilayer membrane of a GUV with a radius $\delta = 20 \text{ \AA}\mu\text{m}$ is $\delta_{\text{tot}} \hat{=} 4\delta^2 / 0.55 \hat{=} 121010$. Thus, the total number of DOGS-NiNTA lipids with fraction $\sim 10\%$ contained in the bilayer NiNTA is $\delta_{\text{tot}} \hat{=} 109$. On the other hand, the total number of $1 \text{ \AA}\mu\text{M}$ protein encapsulated within a $1 \hat{=} 4 \text{ M}$ GUV with $\delta = 20 \text{ \AA}\mu\text{m}$ is $\delta_{\text{prot}} \hat{=} 107$. Thus, complete binding of the protein and DOGS-NiNTA lipids at a 1:1 ratio leaves 102 times more available binding sites. However, the binding of the protein to the membrane may be limited by the size of the protein itself. For example, mDia1 with a molecular weight of 178 kDa has an approximate radius $\delta_{\text{mDia1}} \hat{=} 4 \text{ nm}$, estimated based on molecular weight (Erickson, H. P., Biol. Proced. Online 11, 32-51 (2009)). Thus, the

maximum number of mDia1 that can be bound to the bilayer membrane is estimated to be $\frac{1}{2} \frac{A_{\text{membrane}}}{A_{\text{mDia1}}} \approx 108$. Notably, this is still below the total number of $1 \mu\text{M}$ protein bound to the membrane estimated above ($\frac{1}{2} \frac{A_{\text{membrane}}}{A_{\text{VCA}}} \approx 107$). In the case of VCA with a molecular weight of $\sim 43 \text{ kDa}$ and $\frac{1}{2} \frac{A_{\text{membrane}}}{A_{\text{VCA}}} \approx 108$. Thus, in our experimental conditions, all the his-tagged proteins can be bound to the membrane. We have added this estimation in the Methods section to clarify the experimental condition (pp. 16, lines 505-517) and mentioned in the Results section (pp. 4, lines 102-104).

Minor comments:

1. In my opinion the title is too far-fetched because the authors study GUVs and no biomimetic model cells. Thus, I suggest to change the title accordingly. We have changed the title to be "cell model" instead of "model cells", by which we believe that the title become less far-fetched. We would like to retain this expression to effectively convey the concept of our methodology for creating an in vitro model of cells.

2. Typo l. 65 unilamellar; Typo l.165 These. We have corrected these typos accordingly.

13

Response to Reviewer #3

The manuscript "Composite branched and linear F-actin maximize myosin-induced membrane shape changes in biomimetic model cells" by Sakamoto and Murrell describes membrane shape deformations of liposomes induced by different types of actin-myosin networks that are reconstituted within. The study shows very nicely that a mixture of branched (Arp2/3) and straight (mDia1) actin filaments provide the best substrate for myosin II motors to induce large scale membrane deformations that can be easily detected using confocal fluorescence microscopy. Quantitative image analysis is elegantly used to determine the cortical strain and compute forces and energies that lead to the observed deformations.

This work is a nice continuation of the pioneering work on reconstituted actin-myosin networks of the Murrell lab and provides important novel insights for the community how to realize reconstituted actin networks that enable GUV deformation.

We are grateful to the reviewer for identifying our work as a nice continuation of the pioneering work on reconstituted actin-myosin networks of the Murrell lab and provides important novel insights for the community. We made the following revisions according to the reviewer's comments:

Major points:

1. From the movies it seems that contractions seem to concentrate on one or two locations, conveniently at the equatorial plane. Were GUVs observed/ tested for contractions elsewhere? Or is the blebbistatin inactivation limited to the equatorial plane?

We appreciate this insightful comment. Since we used a standard spinning disk confocal microscope with light exposure throughout the z-direction, the inactivation of blebbistatin was performed both within and outside the focal plane. The deformation may appear to occur in one or two locations by chance (in Movie S6 and S7), possibly due to potential heterogeneity in the distribution of actin and myosin. To clarify this point, we have imaged the liposomes outside of the midplane after the light activation of the midplane (Supplementary Fig. 5a-c) and confirmed that membrane deformation also occurred outside the midplane. In the maximum z-projection of the z-stack images, multiple contracting aster-like structures were formed on the surface of the post-light activated liposomes (Supplementary Fig. 5d). We have clarified these points in the Results section (pp. 6, lines 151-156).

14

a b c

0 min 2 min

Membrane

Midplane
Midplane The other plane The other plane

0 min 2 min
Actin

Midplane
Midplane The other plane The other plane
d
Actin, max. z-projection

2. The authors limit their analysis to GUVs larger than 20 microns. Could they comment on the fate of smaller GUVs? Are they more likely to simply collapse upon myosin activation? Thank you for this comment. We observed that the smaller liposomes tended to exhibit minor membrane deformation. As an example, we added an additional analysis on the membrane strain for liposomes with $\delta \hat{=} 10 \hat{=} \mu\text{m}$, which was significantly smaller than that for liposomes larger than $\delta \hat{=} 20 \hat{=} \mu\text{m}$ (Supplementary Fig. 8). This behavior could be explained by the bending energy cost for the smaller liposomes being much larger than that for the larger ones. Based on the mechanical energy of deformation described in the main text, the ratio of the bending energy cost to the stretching energy cost of the actin cortex scales with radius

15 (curvature), $\delta^{\text{bend}} / \delta^{\text{stretch}} \hat{=} \frac{1}{2} \delta^2$, when the deformation size (correlation length) is comparable to the liposome size, $\delta_c \hat{=} \delta$ (Ito et al., Phys. Rev. E 92, 062711 (2015)). Thus, for the smaller liposomes, the bending energy cost dominates, making it difficult to deform the membrane. In addition to this theoretical consideration, experimentally, the smaller liposomes would have less membrane-localized myosin due to the larger surface-to-volume ratio, which decreases the net active stress applied to the actin cortex. Together, these contributions may limit the extent of deformation for the smaller liposomes. To observe significant membrane deformation sufficient for deformation analysis, we analyzed liposomes larger than $\delta \hat{=} 20 \hat{=} \mu\text{m}$. We have added these points to the main text to clarify the fate of smaller liposomes and the possible mechanisms behind it in the Results section (pp. 9, lines 259-270).

a 0 min 2 min
b
Membrane

Max. memb. strain, ***

0 min 2 min
Actin

$\hat{=} \mu\text{m}$ $\hat{=} \mu\text{m}$

3. Reproducibility: some measurements, e.g. Fig 3e, f, h; Fig 4 c-h; sup fig 1a; sup fig 3 c-e; sup fig 4 a-e, are based on one-time experiments (N=1) without an independent repeat. I am happy to believe that these results are reproducible, however, I think that results shown in a publication should be based on multiple (at least two) independent experiments.

We appreciate this suggestion. We have performed additional experiments or included existing independent experimental data to ensure that all figures, including Fig. 1e-h, Fig. 3e-i, Fig. 4c-h, Fig. 5, Supplementary Fig. 1a, Supplementary Fig. 6, Supplementary Fig. 9, and Supplementary Fig. 10, now present results from at least two independent experiments for all figures. This strengthens the evidence for the reproducibility of our findings.

16

4. Polarized actin flow: The authors mention having observed polarized actin flow in 12 % of liposomes (sup fig. 2e). To make this comparison, one would assume to assess the ratio of GUVs with actin flow compared to the whole number of GUVs that were tested in each experiment. But the figure legend mentions N = 2 for the case of non-polarized GUVs and N=4 for polarized. Or is this a confusion due to similar terms for different things, i.e. the authors compare GUVs prepared with a non-polarized (Arp2/3) and a polarized (Arp2/3 + mDia1) actin network, and ask then, how many of these GUVs show polarized flow?

Thank you for this comment. It was a typo, and the reviewer is right that we meant the ratio of GUVs with actin flow compared to the total number of GUVs tested in each experiment. We agree the previous expression of the caption was confusing. We have revised the caption of Supplementary Fig. 2e to: (e) Frequency of liposomes, calculated as the ratio of liposomes with actin flow compared to the total number of liposomes tested in each experiment (n=102 non-polarized and n=13 polarized, N=6 independent experiments). (Supplementary Fig. 3e).

5. It is mentioned in the methods that the buffer within the GUVs contains 5 mM ATP and an ATP regeneration system. The actin network shows immediate network contraction upon light activation. However, myosin activity seems to cease within the first three minutes. Given the high ATP concentration, one could imagine that myosin activity would be sustained for longer periods. Can it be that the long time of 405 nm light illumination also leads to photodamage of myosin? Can the authors comment on the levels of ATP within the GUV after actin polymerization?

We appreciate this comment. As the reviewer mentioned, there are no contraction dynamics several minutes after light activation in many cases. We provide several comments regarding this point. First, it is well established that the contraction of the actomyosin network finishes after forming contracted asters/aggregates without observable dynamics in two-dimensional actomyosin contraction assays (Murrell & Gardel, PNAS 109, 20820 (2012); Linsmeier et al., Nat. Commun. 7, 12615 (2016); Muresan et al., Nat Commun. 13, 7008 (2022)). In cases where there is no significant actin turnover, it is commonly observed that contraction dynamics cease after a few minutes, eventually forming local contraction spots.

Second, as we discussed in the theoretical model in the Results section, there may be no deformation dynamics when the energetic cost of deformation is balanced by the myosin-generated active stress at some point.

Third, it is possible that photodamage may inactivate the contraction. However, at lower myosin concentrations, we observed persistent contraction for longer periods with slower dynamics compared to the high myosin concentration (Movie S1). Therefore, the possibility of photodamage to myosin can be ruled out.

Finally, the possibility of ATP depletion can be ruled out since the ATP hydrolysis rate of actin is $\sim 4 \mu\text{M min}^{-1}$ at the present actin concentration ($6 \mu\text{M}$) as demonstrated in our recent

publication (Sakamoto & Murrell, Nat. Commun. 15, 3444 (2024)). Thus, the ATP concentration within the liposome remained at $\sim 5 \text{ mM}$ after GUV preparation and actin polymerization ($\sim 10 \text{ min}$). We have included these points in the results section to clarify the potential reasons behind the cessation of network contraction a few minutes after the light activation of myosin (pp. 6, lines 158-167).

6. In addition to that, can the authors comment on the efficacy of myosin incorporation with their approach compared to other encapsulation techniques?

We appreciate this comment. The previous reports suggested that the efficiency of protein incorporation varies with the encapsulation method, such as cDICE, with different studies employing specific protocols involving varied concentrations/contents, buffer compositions, and lipid compositions (Van de Cauter et al., Small Methods 7, e2300416 (2023)).

Unfortunately, assessing encapsulation efficiency successfully is an extensive project beyond the scope of this work. Nevertheless, we have demonstrated rapid actomyosin contraction within liposomes with contraction timescales of minutes, comparable to two-dimensional assays (Murrell & Gardel, PNAS 109, 20820 (2012); Muresan et al., Nat Commun. 13, 7008

(2022)), in a precisely controlled manner depending on myosin concentration. Thus, our system has established robust and efficient myosin incorporation without losing its activity. We have added this point in the Methods section with citation of Van de Caeter et al. (pp. 16, lines 497-503).

Minor points:

1. Page 10, line 295: write 0.5 instead of $\hat{A}^{1/2}$ to be consistent with the text above.

We have corrected this point accordingly.

2. Refs 35 and 49 are the same!

We have corrected this point accordingly.

3. Fig 5a: the phase diagram might be easier to read when adding a third dimension that describes $[mDia1]/[Arp2/3]$ ratio.

Thank you for your suggestion. However, we prefer to keep our plot in 2D as our aim is to directly compare the two experimental readouts of force generation (actin polarity) and transmission (membrane deformation). The overlapping points in the bottom left are acceptable as they are, as they all fall within the non-contractile regime. Also, it is difficult to display $mDia1$ only data in a three-dimensional phase diagram if we add $[mDia1]/[Arp2/3]$ axis, since it will be $[mDia1]/[Arp2/3]=1/0$.

18

4. PIV: please provide the parameters used for PIV analysis, i.e. interrogation box size (single or multiple) and step size.

We have added the PIV parameters in the Methods section (pp. 17, lines 550-554).

5. Sup Fig 2: a-c: it is a bit surprising to see the width of actin flow in the PIV analysis considering the actin is supposed to move along the membrane. Could the authors comment on the reason for this? Are the parameters set too sensitive? Is this due to the overall membrane deformation that occurs within the first minute?

We appreciate this insightful comment. It should be noted that although actin fluorescence is localized, the fluorescence intensity leaks/broadens around the center of the cortex. Therefore, the PIV may result in a wider vector field around the cortex. Additionally, PIV calculates the correlation within a PIV window, which may propagate over adjacent windows. In the additional analysis, we confirmed that a different PIV analysis with a wider width (16 px to 32 px) yielded similar vector fields (Supplementary Fig. 4a-c). Additionally, another example of a liposome showing cortical flow was added (Movie S9).

Also, as the reviewer suggested, it is true that the displacement/deformation of the membrane could affect the PIV of the actin flows. To eliminate the influence of membrane deformation-induced displacement, we performed the PIV analysis on the membrane channel, yielding $\delta@memb$, which was subtracted from the PIV vectors of the actin fluorescence, $\delta@act$. The resulting relative displacement field, $\delta@act \hat{=} \delta@memb$, should reflect the displacement of actin flow alone. As a result, we still observed the flow vectors toward the liposome rear (Supplementary Fig. 4). Therefore, this analysis clarifies the presence of actin flow.

6. f-h: it seems colors for polarized and non-polarized were switched between panels.

We have corrected this point accordingly.

Version 1:

Reviewer comments:

Reviewer #1

(Remarks to the Author)

I am very satisfied with both the answers provided by the authors and the updated version of the manuscript. A lot of in depth explanations have been added in the results sections, and the discussion now addresses important aspects that were not present in the initial submission.

I congratulate the authors on this fine work.

Reviewer #2

(Remarks to the Author)

The authors have responded well to my comments and questions. I recommend their work for acceptance in Communications Biology.

I would only ask them to change l. 489 in the revised manuscript, where they claim that the inverted emulsion technique does not lead to proteins being present in the outer aqueous phase of the GUV-containing solution. Some or many emulsions always burst at the water-oil interface leading to the proteins being present in the outer aqueous phase. Nonetheless, their argument that the cortex cannot be formed on the outside is valid.

Reviewer #3

(Remarks to the Author)

The authors have addressed all points raised by the reviewers and have provided convincing additional data that strengthens their study. This work will be an important contribution to the community working on biomimetic cell model systems and beyond.
